# Dual-windowed Vision Transformer with Angular Self-Attention

## Abstract

Following the great success in natural language processing, transformer-based models have emerged as the competitive model against the convolutional neural networks in computer vision. Vision transformer (ViT) and its subsequent variants have exhibited promising performance in tasks such as image classification, object detection and semantic segmentation. The core of vision transformers is the self-attention mechanism, which models the long-range dependency of different tokens. Conventionally, the attention matrix in self-attention is calculated by the scaled dot-product of *query* (Q) and *key* (K). In this case, the attention weight would depend on norm of Q and K as well as the angle between them. In this paper, we propose a new attention mechanism named angular self-attention, which replaces the scaled dot-product operation with the angular function in order to effectively model the relationship between tokens. In particular, we propose two forms of functions: quadratic and cosine functions, for our angular self-attention. Based on angular self-attention, we design a new vision transformer architecture called dual-windowed angular vision transformer (**DWAViT**). DWAViT is a hierarchical-structured model characterized by the angular self-attention and a new local window mechanism and the new model is supposed to achieve competitive performance on the downstream tasks. We evaluate DWAViT on multiple computer vision benchmark , including image classification on ImageNet-1K, object detection on COCO, and semantic segmentation on ADE20K. Our experimental results also suggest that though our model can achieve promising performance on the tasks, the computational cost of our model is higher than that of the baseline models (i.e., Swin Transformer) due to the new formulation of the self-attention.

## 1 Introduction

Vision transformers have received tremendous attention since its emergence. Inspired by the success of the transformer (Vaswani et al., 2017) in the sequence modeling, Dosovitskiy et al. (Dosovitskiy et al., 2021) proposed the initial architecture of vision transformer which can be regarded as the encoder part of the original transformer (Vaswani et al., 2017). Compared to convolutional neural networks (CNNs), the vision transformer is featured by its ability to transform the spatial visual representation learning on the image into the token-to-token learning, by partitioning the image into multiple patches. Benefited from the ability of the self-attention mechanism that can model the long-range dependence of tokens in the image, vision transformers exhibit on par or better performance against CNNs in many computer vision tasks, such as image classification (Dosovitskiy et al., 2021; Dong et al., 2022), object detection (Carion et al., 2020; He & Todorovic, 2022; Zhang et al., 2022), and semantic segmentation (Zheng et al., 2021; Xie et al., 2021). Despite the merits mentioned above, the shortcoming of vision transformer is also obvious. The low level of the inductive bias requires more large datasets such as Image21K (Deng et al., 2009) and JFT300M (Sun et al., 2017) for model training. Besides, the time complexity of the computation of the self-attention is quadratic to the number of input tokens, which prohibits the application of vision transformers on tasks involving high-resolution images.

To deal with the excessive computation of the self-attention, the subsequent work (Liu et al., 2021; Dong et al., 2022; Huang et al., 2019; Wang et al., 2020; Xia et al., 2022) proposed different local-window mechanisms

to restrict the computation of self-attention in a local window. For instance, the pioneering work Swin Transformer (Liu et al., 2021) adopts the shift windows to reduce the workload of computing self-attention and to facilitate the interaction of local windows. CSwin (Dong et al., 2022) proposes the cross-shaped window, in which the image is split into the horizontal and vertical strips in parallel. Another work (Xia et al., 2022) presents flexible local window, which could be implemented in a data-dependent way.

Another branch of work (Qin et al.; Katharopoulos et al., 2020; Peng et al.; Choromanski et al.) focuses on the in-depth understanding of the self-attention mechanism and proposes new formulations to calculate the attention scores between different pairs of tokens. From the perspective of kernel learning, the interaction of the query and key can be modeled by specific kernel function, and the scaled dot-product operation can be replaced by the softmax-free operation in the self-attention. Usually, the softmax-free operation can lower the time complexity of the computation in self-attention.

In this paper, we present new designs on the local window mechanism and the operation in self-attention. In terms of the local window, we propose a dual window mechanism. As shown in Fig 1, similar to Swin Transformer (Liu et al., 2021), the local window is also imposed on the feature maps for the purpose of reduction of the time complexity. However, unlike

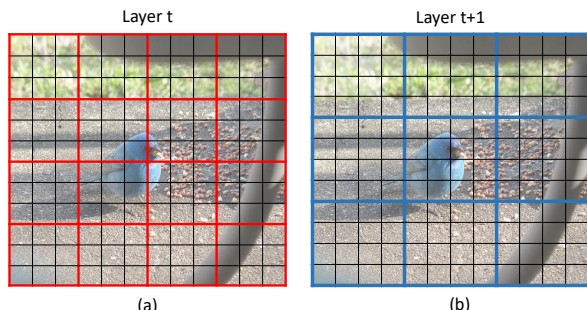

Figure 1: The illustration of the dual window mechanism. The image is partitioned into (a) even number of local windows and (b) odd number of local windows in two layers, respectively. The size of the local window is flexible and the tokens lie on the border of one local window would reside in the interior of the local window in the following layer. The connection of the local window in each layer can be bridged by the operation of the local window in the next layer.

the previous work in which the size of the local window is fixed, the size of the our proposed local window is flexible and is adjustable according to the size of the feature maps. Besides, to mitigate the problem of lacking connections between local windows, the number of local windows is different at layer $t$ and layer $t+1$. For instance, there are even number of the local windows at layer $t$ but odd number of local windows at layer $t+1$. In this case, the tokens that lie in one local window from the first feature map would belong to another local window in the following features map. Since the features maps are partitioned into different number of local windows, the coordinates of the local windows in the adjacent feature maps are different. The tokens in the overlapping area of local windows can bridge the connection of local windows since these tokens would participate in the self-attention calculation within each local windows. With the dynamic interaction of the local windows, the receptive field can be enlarged implicitly, and the ability to model long range relations can also be enhanced.

In traditional self-attention mechanism, the similarity of the query and key is computed by the scaled dot-product. Thus, the similarity would depend on the norm of query and key as well as the angle between them. Inspired by previous work (Wang et al., 2018; Zhao et al., 2020), we notice that scaled dot-product function is not the only choice to model the relationship of tokens. In this paper, we propose the angular self-attention, in which the similarity of query and key is only dependent on the angle between them. To reduce the impact of the norm of the query and key on the relation of tokens, the query and key are L2-normalized, and query and key are distributed on the unit sphere. The relationship of query and key would be determined by the angle between them, and smaller angle could yield larger attention score between a pair of query and key. In angular self-attention, we adopt two forms of functions: quadratic and cosine functions, to model this relationship, and the similarity is further enlarged by the temperature scaling. Our experiments show that the angular self-attention can serve as the competitive alternative for the traditional scaled dot-product self-attention.

Jointly combining the dual window mechanism and angular self-attention, we propose a novel hierarchical-structured vision transformer backbone called dual-windowed angular vision transformer (DWAViT). In DWAViT, the attention score for each pair of query and key is modeled by the temperature-scaled quadratic/cosine functions, and experimental results validate that our quadratic/cosine functions are effective in modeling the relationship between tokens. Besides, dual window mechanism is also adopted in our new

backbone. The feature maps are partitioned into even/odd number of local windows in the layers of DWAViT alternatively. The dual window mechanism can preserve the ability to model long-range relationship between tokens. However, due to the new formulation of the self-attention. the computational cost of our model is higher than that of baseline models with a similar size (i.e., Swin Transformer (Liu et al., 2021)). With proper partition of the feature maps, our DWAViT can be applied to the downstream tasks (e.g., object detection, semantic segmentation) which involve high-resolution images. The major contributions of this paper are as follows:

- We propose the dual window mechanism to split the global feature map into a couple of smaller localized feature maps. By partitioning the feature maps in even/odd number of local windows in an alternative way, the lack of connection between local windows can be alleviated.

- We propose the angular self-attention in which the scaled dot-product operation is replaced by the temperature-scaled quadratic/cosine functions. Our proposed angular self-attention can model the long range relationship of tokens and is the competitive alternative for the traditional scaled dot-product self-attention.

- The dual-windowed angular vision transformer (DWAViT) is proposed by jointly combining the dual window and angular self-attention. The DWAViT is evaluated on a series of dense prediction tasks and achieve competitive performance on ImageNet image classification, COCO object detection, and ADE20K semantic segmentation.

## 2 Related Work

**Vision Transformers**. The pioneering work (Parmar et al., 2018; Wang et al., 2018) first introduced the self-attention mechanism to the computer vision field and some early work Ramachandran et al. (2019); Cordonnier et al. (2019) applied self-attention in the computer vision tasks. Dosovitskiy et al. (Dosovitskiy et al., 2021) proposed the transformer-based backbone architecture called vision transformers (ViTs) (Dosovitskiy et al., 2021). With the new paradigm of representative learning, ViTs achieve on par or better performance on image classification, object detection and semantic segmentation against CNNs. Since the emergence of vision transformers, plenty of work (Touvron et al., 2021a; 2022) has been done on this field and the subsequent work aims to improve the ViTs on different aspects. DeiT (Touvron et al., 2021a; 2022) proposes new training recipe to reduce the high demand of ViTs for the very large datasets. With the techniques provided by DeiT (Touvron et al., 2021a; 2022), ViTs can pretrained from scratch on smaller datasets such as ImageNet-1K (Deng et al., 2009) compared to Image21K (Deng et al., 2009) and JFT300M (Sun et al., 2017). Besides, ViTs also borrow the idea form the modern CNN architectures (He et al., 2016; Howard et al., 2017; Sandler et al., 2018; Tan & Le, 2019; 2021; 2019; Huang et al., 2017; Liu et al., 2022b; Rao et al.; Wang et al., 2022a; Dai et al., 2021) to improve the ability of representative learning and develop hierarchical pyramid structure to handle the multi-scale feature maps. The pyramid-structured ViTs usually have four stage and in each stage the size of the feature maps is half of that in the previous stage while the dimension is doubled. Another line of work (Wu et al., 2021; Guo et al., 2022; Xiao et al., 2021; Tu et al., 2022; Yuan et al., 2021; Srinivas et al., 2021; Chen et al., 2022; Mehta & Rastegari; Peng et al., 2021) incorporates the convolution operation into the architecture of the vision transformers at different location. The performance of the hybrid vision transformers are further improved by fusing the local information learned by CNNs and global dependence information obtained by self-attention. To mitigate the computational cost of the global self-attention which is quadratic to the size of the input features. Some work (Lee et al., 2022; Chen et al., 2021) learn the contextual information from the multi-scale patch embedding. An extensive work (Dong et al., 2022; Liu et al., 2022a; 2021; Xia et al., 2022; Wang et al., 2020; Hassani et al., 2022; Han et al., 2021; Huang et al., 2019; Ren et al., 2022) proposes different local window mechanism to reduce the computational cost. The self-attention is performed within the local windows and the connection of different local window is achieved by some techniques such as shifted window (Liu et al., 2021) or cross-shaped window (Dong et al., 2022). In our paper we propose a new local window mechanism called dual window and the connection of local window can be achieved in a simple way.

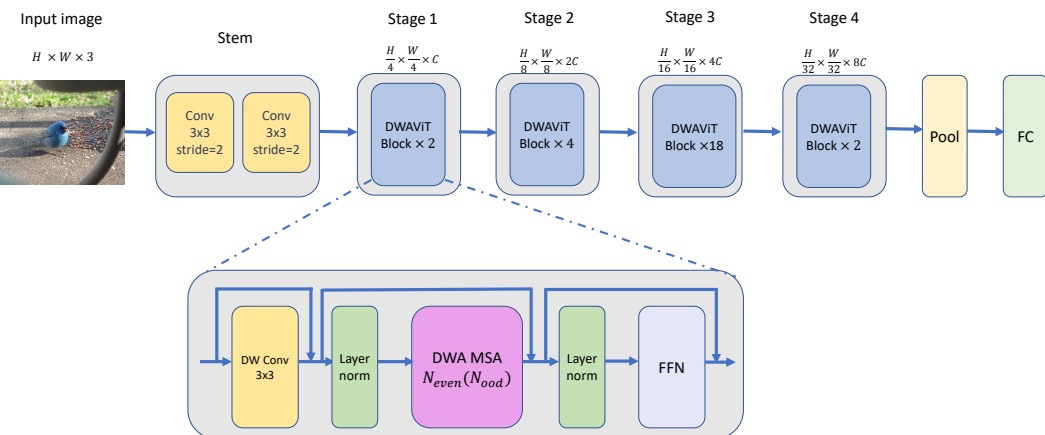

Figure 2: The illustration of our proposed dual-windowed angular vision transformer (DWAViT). Similar to previous work our backbone adopts the hierarchical pyramid structure. The core module in our backbone is the dual-windowed angular multi-head self-attention (DWA MSA) which jointly combines the dual window mechanism and angular self-attention. In each block the feature maps are divided into even/odd number of local windows. Besides. The depthwise convolution provides the conditional positional embedding.

**Self-attention**. Apart from the traditional scaled dot-product self-attention, different forms of self-attention mechanism are also proposed. Early work (Wang et al., 2018; Zhao et al., 2020) explored the general form of the function in the self-attention and proposed several operations such as dot-product and concatenation. XciT (Ali et al., 2021) proposed cross-covariance attention (XCA) in which the attention is performed on g over channels instead of tokens. MaxVit (Tu et al., 2022) and DaViT (Ding et al., 2022) proposed grid attention and channel group attention, respectively. These attentions are also performed on channels dimension rather than spatial dimension. To reduce the the computational cost, the efficient self-attention is proposed to approximate the traditional softmax self-attention under the lens of kernel learning. Linear transformer (Katharopoulos et al., 2020) suggests that softmax function can be removed and the similarity of tokens can be obtained by pure dot product of query and key. RFA (Peng et al.) and performer (Choromanski et al.) approximate the softmax attention with positive random features. CosFormer (Qin et al.) proposed cos-based re-weighting self-attention in which the attention score is calculated by the weighted dot-product of query and key. In SOFT (Lu et al., 2021), the dot-product similarity is replaced by the Gaussian kernel function. In our paper, we also propose a new self-attention mechanism called angular self-attention, in which the similarity of tokens is calculated from a quadratic function.

## 3 Methodology

### 3.1 Dual Window

As shown if Fig 1, the feature maps is partitioned into even number of local windows and odd number of local window at layer t and layer t+1 alternatively. The feature maps is padded if necessary. Suppose the original size of feature map is $h \times w$. after the padding, the size of the padded feature map is $h' \times w'$. The number of the local window is $N_{even} = n_{even}^2 (N_{ood} = n_{ood}^2)$. $n_{even}(n_{ood})$ is the number of local window per side. Thus, the size of local window is $\frac{h}{n_{even}} \times \frac{w}{n_{even}} (\frac{h}{n_{ood}} \times \frac{w}{n_{ood}})$. Compared to Swin Transformer (Liu et al., 2021) that bridge the connection of different local windows by complicated techniques such as cycle shift, we solve this problem in a simple way. Notice that tokens lie on the border of one local window would reside in the interior of the local window in the following layer. Therefore, the tokens on the border of the local window at one layer can participate in the self-attention calculation with the tokens from other local windows in the next layer. The dynamic interaction of the local windows can facilitate the propagation of the information between local windows, The actual size of receptive field would be larger than the size of local window and the ability to model the long-range relationship of tokens would also be enhanced.

### 3.2 Angular Self-Attention

Self-attention can be regarded as a weighted combination of the input sequence, where the weights are determined by the similarities between elements of the input sequence. We use $\mathcal{O}_i \in \mathcal{R}^d$ to denote the generated embedding of token $i$ from self-attention. Then the general form of self-attention could be written as:

$$\mathcal{O}_i = \sum_j \frac{\mathcal{S}(Q_i, K_j)}{\sum_j \mathcal{S}(Q_i, K_j)} V_j, \tag{1}$$

where $\mathcal{S}(\cdot)$ represents the similarities between $Q$ and $K$ and it has many forms according to the previous work (Wang et al., 2018; Zhao et al., 2020). if $\mathcal{S}(Q_i, K_j) = \exp(Q_i \cdot K_j / \sqrt{d_k})$, Eq. 1 would become the scaled dot-product attention as we commonly see in vision transformers. The formulation of scaled dot-product self-attention in vision transformer is:

$$\text{Attention}(Q, K, V) = \text{Softmax}(\frac{QK^T}{\sqrt{d_k}})V. \tag{2}$$

In dot-product self-attention, the attention weight is generated from the scaled dot-product between Q and K. The dot-product of $Q_i$ and $K_j$ can be expanded as $Q_i \cdot K_j = ||Q_i|| ||K_j|| \cos\theta$. It indicates that the similarity would depend on the L2 norm of $Q$ and $K$ as well as their angles $\theta$. In our paper, we propose angular self-attention, in which we use angular function $s(\theta)$ to replace the conventional scaled dot-product operation. Then the self-attention could be reformulated as:

$$\mathcal{O}_i = \sum_j \frac{\exp(s(\theta_{ij})/\tau)}{\sum_j \exp(s(\theta_{ij})/\tau)} V_j, \tag{3}$$

where $\theta_{ij} = \arccos(\hat{Q}_i \cdot \hat{K}_j)$, $\hat{Q}$ and $\hat{K}$ are L2 normalized query and key, respectively. $\tau$ is the temperature hyper-parameter that regulates the attention weight of each token.

When Q and K are normalized, they can be distributed on the surface of the unit sphere. Then the attention weight obtained from our angular self-attention is solely dependent on the angle $\theta$. Through our training the angles $\theta$ between different Q and K would be adjusted to model the relationship of different tokens and make the vision transformer achieve strong representative ability. Thus, we propose two alternative functions for $s(\theta)$ in Eq. 3. They are cosine function $s(\theta) = \cos(\theta)$ and quadratic function $s(\theta) = 1 - \frac{4\theta^2}{\pi^2}$. In angular self-attention, the similarity of $Q$ and $K$ would solely depend on their angles. The matrix form of angular self-attention could be formulated as:

$$\text{Attention}(Q, K, V) = \text{Softmax}(\frac{\hat{Q}\hat{K}}{\tau})V$$
$$\text{Attention}(Q, K, V) = \text{Softmax}(\frac{1 - 4\Theta^2/\pi^2}{\tau})V, \tag{4}$$

where $\Theta = \arccos(\hat{Q} \cdot \hat{K})$. $\hat{Q}$ and $\hat{K}$ are L2 normalized query and key, respectively.

The cosine similarity and quadratic distance has common mathematical properties. Both functions are descending when $\theta \in [0, \pi]$, which means that the tokes with larger angles would have less weaker relationships. Specifically, when $\theta \in [0, \pi/2]$, $\cos\theta \approx 1 - \theta^2/2 \approx 1 - 4\theta^2/\pi^2$, when $\theta \in (\pi/2, \pi]$, $1 - 4\theta^2/\pi^2 < \cos\theta < 0$, which means that tokens with angles larger than $\pi/2$ have weaker relationships in quadratic function than that of in cos function. Our experiments suggest that in most tasks like image classification and object detection, the performance of quadratic and cosine functions is comparable and the difference is very slight ($<0.5\%$). However, in semantic segmentation, the performance of cosine function is better than that of quadratic function ($>1.0\%$).

### 3.3 Overall Architecture

We replace the traditional scaled dot-product self-attention with our angular self-attention, and integrate the dual window mechanism to build our dual-windowed angular vision transformer (DWAViT). The overall

Table 1: The details of the DWAViT variants.

| Models | #Dim | #Blocks | #Heads | #Param(M) | #FLOPs(G) |
|---|---|---|---|---|---|
| DWAViT-Tiny | [64,128,256,512] | [2,4,18,2] | [1,2,4,8] | 22.7 | 4.2 |
| DWAViT-Small | [80,160,320,640] | [3,6,21,3] | [1,2,4,8] | 44.6 | 8.2 |
| DWAViT-Base | [96,192,384,768] | [4,8,24,4] | [1,2,4,8] | 77.4 | 14.3 |

illustration of our proposed dual-windowed angular vision transformer (DWAViT) is illustrated in Fig 2. Similar to the previous work (Wang et al., 2021; 2022b; Ding et al., 2022; Fan et al., 2021; Li et al., 2022; Liu et al., 2021; 2022a), the DWAViT also adopt the hierarchical pyramid structure that take advantage of the multi-scale resolution of feature maps for the dense prediction task. The size of the input image is $H \times W \times 3$. Instead of adopting the convolutional layer with large kernel, we follow the work (Xiao et al., 2021) and leverage the two stacked convolutional layer as the stem to generate patch embedding. For each convolutional layer, the kernel size is $3 \times 3$ and the stride is $2 \times 2$. The size of the output from the stem is $\frac{H}{4} \times \frac{W}{4} \times C$.

The DWAViT consists of four stages in which the size of the feature maps is half of that from previous stage while the dimension is doubled compared to that from previous stage. Between two adjacent stage we adopt a convolutional layer with kernel size of $2 \times 2$ and stride of 2 to downsample the feature maps. Each stage consists of multiple blocks which include the depthwise convolution (Chollet, 2017) that generates the conditional positional embedding (CPE) (Chu et al., 2021b) , the dual-windowed angular multi-head self-attention (DWA MSA) and feed-forward network (FFN). Compared to absolute positional embedding (APE) (Vaswani et al., 2017) that could only provide the the positional information for the fixed length of sequence, The CPE can provide flexible positional information adaptive to various length of input sequence that is often seen in the downstream tasks. Relative positional embedding (RPE) (Liu et al., 2021; Shaw et al., 2018) provide the relative positional information within the window. However, since the size of window is different in each stage, we don't adopt the RPE in our DWAViT.

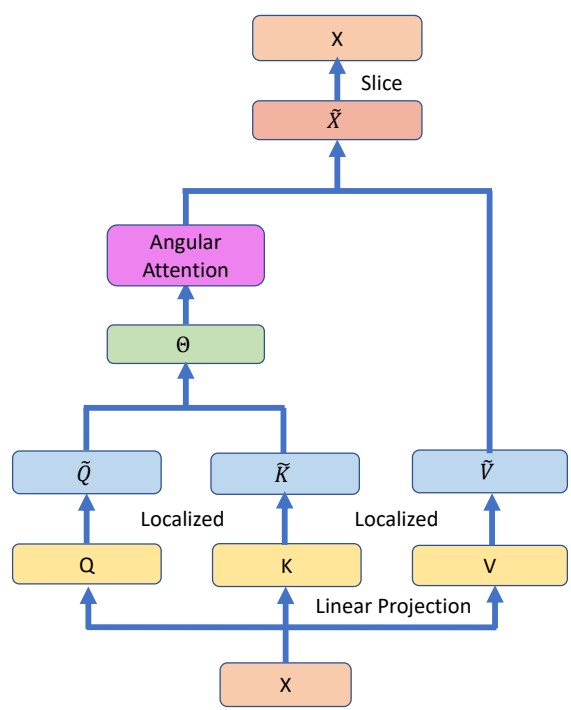

Figure 3: The illustration of the pipeline in the dual-windowed angular multi-head self-attention (DWA MSA). The Q, K and V are localized by dividing them into a couple of local windows. The traditional scaled dot-product operation is replaced by the temperature-scaled angular function in the calculation of attention matrix

The dual-windowed angular multi-head self-attention (DWA MSA) serves the core function for our backbone. It jointly combines the dual window mechanism and angular self-attention. The details of DWA MSA is illustrated in Fig 3, Suppose the input feature is $X \in \mathbb{R}^{h \times w \times D}$, $N = N_{even}$ or $N_{ood}$ is the number of local windows and $n = \sqrt{N}$ is the number of local windows per side. After linear projection, we obtain query, key and value $Q, K, V \in \mathbb{R}^{h \times w \times D}$. Instead of splitting the $X$ into smaller local windows, we split $Q$ and $K$ into $N$ local windows. The size of the local window is $\frac{h'}{n} \times \frac{w'}{n}$. The $h'$ and $w'$ are the height and width of the padded feature maps. In each stage, the partition with even/ood number of windows take turns and the value of $N_{even}(N_{ood})$ vary according to the size of the feature maps. After the partition, for $k^{th}$ head, we obtain the localized query $\tilde{Q}^k = \{Q_i^k, Q_2^k, ..., Q_N^k\}$ and key $\tilde{K}^k = \{K_i^k, K_2^k, ..., K_N^k\}$. the localized query and key are L2 normalized and the angle matrix $\Theta$ is calculated from the normalized query and key. Next the new embedding is calculated from the localized query, key and value by Eq. 4. The new feature maps of each local window is obtained by the concatenation of the embedding from each head

$\tilde{X}_i = \text{concat}(\tilde{X}_i^1, \tilde{X}_i^2, ..., \tilde{X}_i^K)$. $K$ is the number of heads. We concatenate the feature maps of each local window to form the complete feature map $\tilde{X} = \text{concat}(\tilde{X}_1, \tilde{X}_2, ..., \tilde{X}_N)$. The size of $\tilde{X}$ is larger than that of original feature map. To restore it to the original size, the final feature map is obtained from $\tilde{X}$ with slice operation:

$$X = \tilde{X}[\text{top} : h - \text{bottom}, \text{left} : w - \text{right}], \tag{5}$$

where top, bottom, left and right are the size of padding on the top, bottom, left and right of the feature maps, respectively. With all the components aforementioned above, the pipeline of the block in our DWAViT can be formulated as:

$$\begin{aligned} Z^\ell &= \text{DWConv}(X^{\ell-1}) + X^{\ell-1}, \\ \tilde{X}_\ell &= \text{DWA-attention}(\text{LN}(Z^\ell)) + Z^\ell, \\ X_\ell &= \text{FFN}(\text{LN}(\tilde{X}_\ell)) + \tilde{X}_\ell, \end{aligned} \tag{6}$$

where $X^{\ell-1}$ denote the output feature from the $\ell^{th}$ block in the backbone and DWConv denotes the depthwise convolution.

In our DWAViT, there are strong connections between the two key components: angular self-attention and dual local window mechanism. On one hand, The dual window mechanism can confine the operation of self-attention in localized areas. On the other hand, our empirical study suggests that our angular self-attention can achieve better performance with our dual local window techniques than that with previous local window techniques (i.e., Swin Transformer (Liu et al., 2021)). To wrap up, the two proposed techniques (dual windows and angular self-attention) have strong mutual connections and are two indispensable components in our DWAViT.

### 3.4 Architecture Variants

We build three variants of DWAViT with different number of parameters and FLOPs, namely DWAViT-Tiny, DWAViT-Small and DWAViT-Base. For all the variants, the number of local window in each stage is set to $(64,49)$, $(16,9)$, $(4,1)$, $(1,1)$ in image classification, respectively. In stage one, the size of the local window would be $7 \times 7$ and $8 \times 8$ in two consecutive blocks. For the downstream tasks such as object detection and semantic segmentation, since the size of the input image is larger, the number of local windows in DWAViT is also different. The details of three DWAViT variants are illustrated in Table 1.

### 3.5 Time Complexity Analysis

Suppose the original size of feature map is $h \times w$ and the dimension is $C$. After padding the size of feature maps become $h' \times w'$. The total number of local window is $N$. The time complexity of the linear projection is $4hwC^2$. Since the self-attention is performed on the padded feature maps, the time complexity for the self-attention calculation is $2(h'w')^2C/N$. Thus, the total time complexity of our DWA MSA is:

$$\Omega(\text{DWA MSA}) = 4hwC^2 + 2(h'w')^2C/N. \tag{7}$$

As illustrated in Eq. 7, In order to reduce the time complexity, we should choose a large value of $N$ while keeping the $h'$ and $w'$ as close as to the original value. Note that though angular self-attention includes some operations like arccos and L2 normalization, they do not increase the time complexity. However, the memory demand of angular self-attention is larger than that of traditional self-attention.

### 3.6 Theoretical Analysis

As aforementioned, we propose quadratic and cosine functions to model the relationships of tokens. Compared to the scale dot-product function, the difference of our method is that we map the Q and K features on the unit sphere and the relationship of tokens is only dependent on the angles between them. To better understand the angular self-attention, it is essential to investigate the relationship of tokens in our method. Thus, we provided the Proposition to analyze this problem.

**Proposition 1** *Suppose the angles between the query of a token and the keys of all the tokens match* $\theta_0 \leq \theta_1 \leq \cdots \leq \theta_{J-1}$. *$J$ is the total number of the tokens. In angular self-attention, the embedding of the token* $\mathcal{O}_i \propto V_0 + \omega_1 V_1 + \cdots + \omega_{J-1} V_{J-1}$, *where* $\omega_k = \exp(\frac{s(\theta_k)-s(\theta_0)}{\tau}) \in (0,1)$. $s(\theta) = 1 - \frac{4\theta^2}{\pi^2}$ *for quadratic self-attention and* $s(\theta) = \cos\theta$ *for cosine self-attention.*

The proposition states that the embedding of a token can be regarded as the combination of the value vectors with the relative weight denoted by $w_k$. The relative weight of the value vector with smallest angle to the target query is normalized to 1. And the relative weight of a value vector is smaller if the corresponding angle is larger. The proposition suggest the larger contribution of the value vector to the embedding of the target token with smaller angles between them. The proof can be found in Appendix.

## 4  Experiments

In this section, we evaluate our proposed DWAViT on ImageNet-1K (Deng et al., 2009) classification, COCO (Lin et al., 2014) object detection, and ADE20K (Zhou et al., 2017) semantic segmentation. Besides, we also implement the ablation study to investigate the effectiveness of angular self-attention and compare the results of angular self-attention against that of traditional scaled dot-product self-attention on benchmarks.

### 4.1  ImageNet-1K Classification

this experiment we adopt the same training recipe as previous work (Touvron et al., 2021a; Li et al., 2022; Lee et al., 2022; Dong et al., 2022) for fair comparison. The training strategies include repeated data augmentation methods and the EMA (Polyak & Juditsky, 1992). The total training epoch is 300 with the first 20 epochs as warm-up. We adopt the AdamW (Kingma & Ba, 2014) algorithm to optimize the model. The initial learning rate is 1.2e-3 and the weight decay is 0.05. The learning rate is adjusted according to the cosine learning rate schedule. The drop path rate is 0.1 and the input image is resized to $224 \times 224$. The mlp ratio for all the DWAViT variants is set to 4. The number of windows in each stage is (64,49), (16,9), (4,1), (1,1). The temperature in an angular self-attention is 0.1 for DWAViT-T and DWAViT-S and 0.25 for DWAViT-B, respectively. All the experiments are running on NVIDIA A100. Since the model is trained on ImageNet-1K for many training epochs (300 epochs), the variance could be diminished considerably and become insignificant compared to the main results. As a consequence, only the major results are reported.

The results are illustrated in Table 2. Our proposed DWAViT is compared against the previous state-of-the-art vision transformers and CNNs including the CSwin (Dong et al., 2022), MViTv2 (Li et al., 2022), DaViT (Ding et al., 2022) and ConvNeXt (Liu et al., 2022b). Specifically, Swin Transformer (Liu et al., 2021), Focal Transformer (Yang et al., 2021), DaViT (Ding et al., 2022), MViTv2 (Li et al., 2022) and CSWin Transformer (Dong et al., 2022) are baselines for the exact comparison. The experimental results show that under the similar amount of parameters, the DWAViT-T can outperform the latest vision transformers and CNNs, The top-1 accuracy of the DWAViT-T can achieve 82.8%, which is even 0.1% higher than that of CSwin-T (Dong et al., 2022). As for the small-sized model, our DWAViT-S can achieve the top-1 accuracy of 83.6% in the classification task, which is on par with that of CSWin (Dong et al., 2022) and MViTv2 (Li et al., 2022). For base-sized models, our DWAViT-B with cosine self-attention can achieve 83.9% accuracy in classification task.

### 4.2  COCO Object Detection

Next, we evaluate our model on COCO object detection task. The COCO dataset has 118K images for training and 5K images for validation. We adopt the Mask R-CNN (He et al., 2017) and Cascade Mask R-CNN (Cai & Vasconcelos, 2018) as the framework and our DWAViT serves as the backbone. For a fair comparison, we follow the same training recipe as the previous work (Touvron et al., 2021a; Li et al., 2022; Lee et al., 2022; Dong et al., 2022) and perform the experiment with MMDetection toolbox (Chen et al., 2019). In order to tackle with the images with high resolution. The number of local windows in object detection task is different from that in image classification task. The number of window in each stage is (256,225), (64,49), (16,9), (4,1), respectively. In both framework, the size of the local window in each stage

Table 2: The performance of our proposed DWAViT and the baseline models on the ImageNet-1K classification. The resolution of the image is 224 × 224. [cos] and [quad] denote cosine and quadratic function, respectively.

| Model | #Param(M) | #FLOPs(G) | Top-1 Acc |
|---|---|---|---|
| ResNet-50 (He et al., 2016) | 25.0 | 4.1G | 76.2 |
| DeiT-S (Touvron et al., 2021a) | 22.1 | 4.5 | 79.8 |
| PVT-S (Wang et al., 2021) | 24.5 | 3.8 | 79.8 |
| RegNetY-4G (Radosavovic et al., 2020) | 21.0 | 4.0 | 80.0 |
| CrossViT-S (Chen et al., 2021) | 26.7 | 5.6 | 81.0 |
| TNT-S (Han et al., 2021) | 23.8 | 5.2 | 81.3 |
| Swin-T (Liu et al., 2021) | 28.3 | 4.5 | 81.2 |
| CoAtNet-0 (Dai et al., 2021) | 25 | 4.0 | 81.6 |
| CvT-13 (Wu et al., 2021) | 20.0 | 4.5 | 81.6 |
| CaiT–XS-24 (Touvron et al., 2021b) | 26.6 | 5.4 | 81.8 |
| ViL-S (Zhang et al., 2021) | 24.6 | 5.1 | 82.0 |
| PVTv2-B2 (Wang et al., 2022b) | 25.4 | 4.0 | 82.0 |
| ConvNeXt-T (Liu et al., 2022b) | 29 | 5.0 | 82.1 |
| Focal-T (Yang et al., 2021) | 29.1 | 4.9 | 82.2 |
| DaViT-T (Ding et al., 2022) | 28.3 | 4.5 | **82.8** |
| MViTv2-T (Li et al., 2022) | 24 | 4.7 | 82.3 |
| CSWin-T (Dong et al., 2022) | 23 | 4.3 | 82.7 |
| DWAViT-T[cos] (Ours) | 22.7 | 4.2 | 82.7 |
| DWAViT-T[quad] (Ours) | 22.7 | 4.2 | **82.8** |
| ResNet-101 (He et al., 2016) | 45.0 | 7.9 | 77.4 |
| PVT-M (Wang et al., 2021) | 44.2 | 6.7 | 81.2 |
| RegNetY-8G (Radosavovic et al., 2020) | 39.0 | 8.0 | 81.7 |
| Swin-S (Liu et al., 2021) | 49.6 | 8.7 | 83.1 |
| CoAtNet-1 (Dai et al., 2021) | 42.0 | 8 | 83.3 |
| CvT-21 (Wu et al., 2021) | 32.0 | 7.1 | 82.5 |
| ViL-M (Zhang et al., 2021) | 39.7 | 9.1 | 83.3 |
| PVTv2-B (Wang et al., 2022b) | 45.2 | 6.9 | 83.2 |
| ConvNeXt-S (Liu et al., 2022b) | 50.0 | 9.0 | 83.1 |
| Focal-S (Yang et al., 2021) | 51.1 | 9.1 | 83.5 |
| MViTv2-S (Li et al., 2022) | 35 | 7.0 | **83.6** |
| CSWin-S (Dong et al., 2022) | 35 | 6.9 | **83.6** |
| DWAViT-S[cos] (Ours) | 44.6 | 8.2 | 83.5 |
| DWAViT-S[quad] (Ours) | 44.6 | 8.2 | **83.6** |
| ResNet-152 (He et al., 2016) | 60.0 | 11.0 | 78.3 |
| PVT-L (Wang et al., 2021) | 61.4 | 9.8 | 81.7 |
| DeiT-B (Touvron et al., 2021a) | 86.7 | 17.4 | 81.8 |
| Swin-B (Liu et al., 2021) | 87.8 | 15.4 | 83.4 |
| ViL-B (Zhang et al., 2021) | 55.7 | 13.4 | 83.2 |
| Focal-B (Yang et al., 2021) | 89.8 | 16.0 | 83.8 |
| DWAViT-B[cos] (Ours) | 77.4 | 14.3 | **83.9** |
| DWAViT-B[quad] (Ours) | 77.4 | 14.3 | 83.8 |

Table 3: Object detection and instance segmentation performance the our model and the baseline models with Mask R-CNN framework with 1x schedule training scheme. The FLOPs are measured at resolution 800 × 1280. [cos] and [quad] denote cosine and quadratic function, respectively.

| Backbone | #Param(M) | #FLOPs(G) | $AP^b$ | $AP^b_{50}$ | $AP^b_{75}$ | $AP^m$ | $AP^m_{50}$ | $AP^m_{75}$ |
|---|---|---|---|---|---|---|---|---|
| ResNet-50 (He et al., 2016) | 44 | 260 | 38.0 | 58.6 | 41.4 | 34.4 | 55.1 | 36.7 |
| PVT-S (Wang et al., 2021) | 44 | 245 | 40.4 | 62.9 | 43.8 | 37.8 | 60.1 | 40.3 |
| ViL-S (Zhang et al., 2021) | 45 | 218 | 44.9 | 67.1 | 49.3 | 41.0 | 64.2 | 44.1 |
| TwinsP-S (Chu et al., 2021a) | 44 | 245 | 42.9 | 65.8 | 47.1 | 40.0 | 62.7 | 42.9 |
| Twins-S (Chu et al., 2021a) | 44 | 228 | 43.4 | 66.0 | 47.3 | 40.3 | 63.2 | 43.4 |
| Swin-T (Liu et al., 2021) | 48 | 264 | 42.2 | 64.6 | 46.2 | 39.1 | 61.6 | 42.0 |
| DAT-T (Xia et al., 2022) | 48 | 272 | 44.4 | 67.6 | 48.5 | 40.4 | 64.2 | 43.1 |
| CSWin-T (Dong et al., 2022) | 42 | 279 | **46.7** | 68.6 | **51.3** | **42.2** | 65.6 | 45.4 |
| DWAViT-T[cos] (Ours) | 42 | 255 | 46.2 | 69.2 | 50.8 | 41.7 | 65.9 | 44.7 |
| DWAViT-T[quad] (Ours) | 42 | 255 | 46.6 | **69.6** | **51.3** | **42.2** | **66.3** | **45.7** |
| Res101 (He et al., 2016) | 63 | 336 | 40.4 | 61.1 | 44.2 | 36.4 | 57.7 | 38.8 |
| PVT-M (Wang et al., 2021) | 64 | 302 | 42.0 | 64.4 | 45.6 | 39.0 | 61.6 | 42.1 |
| ViL-M (Zhang et al., 2021) | 60 | 261 | 44.6 | 66.3 | 48.5 | 40.7 | 63.8 | 43.7 |
| TwinsP-B (Chu et al., 2021a) | 64 | 302 | 44.6 | 66.7 | 48.9 | 40.9 | 63.8 | 44.2 |
| Twins-B (Chu et al., 2021a) | 76 | 340 | 45.2 | 67.6 | 49.3 | 41.5 | 64.5 | 44.8 |
| Swin-S (Liu et al., 2021) | 69 | 354 | 44.8 | 66.6 | 48.9 | 40.9 | 63.4 | 44.2 |
| CSWin-S (Dong et al., 2022) | 54 | 342 | 47.9 | 70.1 | 52.6 | 43.2 | 67.1 | 46.2 |
| DAT-S (Xia et al., 2022) | 69 | 378 | 47.1 | 69.9 | 51.5 | 42.5 | 66.7 | 45.4 |
| DWAViT-S[cos] (Ours) | 64 | 338 | 48.0 | 70.7 | 52.6 | **43.3** | 67.6 | 46.5 |
| DWAViT-S[quad] (Ours) | 64 | 338 | **48.2** | 70.8 | 52.8 | **43.3** | **67.7** | **46.6** |
| X101-64 (Xie et al., 2017) | 101 | 493 | 42.8 | 63.8 | 47.3 | 38.4 | 60.6 | 41.3 |
| PVT-L (Wang et al., 2021) | 81 | 364 | 42.9 | 65.0 | 46.6 | 39.5 | 61.9 | 42.5 |
| CSWin-B (Dong et al., 2022) | 97 | 526 | **48.7** | 70.4 | **53.9** | **43.9** | 67.8 | **47.3** |
| DWAViT-B[cos] (Ours) | 97 | 462 | 48.6 | 71.1 | 53,7 | 43.6 | **68.0** | 46.9 |
| DWAViT-B[quad] (Ours) | 97 | 462 | 48.6 | **71.2** | 53.6 | 43.6 | 67.9 | 47.1 |

is half of that in the previous stage. We use the model pretrained on ImageNet-1K and fine-tune it on the COCO dataset with 1× and 3 × schedule with 12 and 36 epochs, respectively.

The results on object detection and instance segmentation of our model and the baseline models with Mask R-CNN (He et al., 2017) framework with 1× schedule are illustrated in Table 3. The baseline methods include the latest ViT models such as CSwin (Dong et al., 2022) and DAT (Xia et al., 2022). Specifically, Swin Transformer (Liu et al., 2021), DAT (Xia et al., 2022) and CSwin (Dong et al., 2022) are baselines for exact comparison. These baselines and our method adopt MMDetection toolbox (Chen et al., 2019) to perform the experiment and use models pre-trained on ImageNet-1K only as the feature extractor. For tiny-sized models the experimental results show that the DWAViT-T can achieve on par or better result against that of CSWin (Dong et al., 2022). For instance, the $AP^b_{50}$ of DWAViT-T(quad) can achieve 69.6%, which is 1.0% higher than that of CSWin (Dong et al., 2022). And the $AP^m$ of DWAViT-T(quad) is 42.2%, which is on par with that of CSWin (Dong et al., 2022). The DWAViT-S with quadratic self-attention can outperform all the baseline methods on all the metrics. The $AP^b$ and $AP^m$ can reach 48.2% and 43.3%, respectively. Furthermore, DWAViT-B can achieve best results on $AP^b_{50}$ and $AP^m_{50}$.

The results on object detection and instance segmentation of our our model and the baseline models with Mask R-CNN (He et al., 2017) framework with 3× schedule are illustrated in Table 4. The baseline methods include the latest ViT models such as MViTv2 (Li et al., 2022), DAT (Xia et al., 2022) and DAViT (Xia et al., 2022). Specifically, Swin Transformer (Liu et al., 2021), Focal Transformer (Yang et al., 2021), XciT (Ali et al., 2021), DAT (Xia et al., 2022) and CSwin (Dong et al., 2022) are baselines for exact comparison. The experimental results show that the DWAViT can achieve on par or better result with latest baseline methods. For DWAViT-T, the $AP^b$ can achieve 48.8%, which is 0.4% higher than that of MViTv2-T (Li et al., 2022). the $AP_m$ of DWAViT-T is 43.8%, which is on par with that of MViTv2-T (Li et al., 2022). The DWAViT-S achieves 49.1% on $AP^b$ and 44.4% on $AP^m$, which outperforms all the baseline methods. And the experimental results suggest that our DWAViT-B can outperform the Swin-B (Liu et al., 2021) on this task.

Table 4: Object detection and instance segmentation performance the our model and the baseline models with Mask R-CNN framework. The model is trained with 3x scheme. The FLOPs are measured at resolution $800 \times 1280$. [cos] and [quad] denote cosine and quadratic function, respectively.

| Backbone | #Param(M) | #FLOPs(G) | $AP^b$ | $AP_{50}^b$ | $AP_{75}^b$ | $AP^m$ | $AP_{50}^m$ | $AP_{75}^m$ |
|---|---|---|---|---|---|---|---|---|
| ResNet-50 (He et al., 2016) | 44 | 260 | 41.0 | 61.7 | 44.9 | 37.1 | 58.4 | 40.1 |
| ConvNeXt-T (Liu et al., 2022b) | 48 | 262 | 46.2 | 67.9 | 50.8 | 41.7 | 65.0 | 44.9 |
| PVT-S (Wang et al., 2021) | 44 | 245 | 43.0 | 65.3 | 46.9 | 39.9 | 62.5 | 42.8 |
| ViL-S (Zhang et al., 2021) | 45 | 218 | 47.1 | 68.7 | 51.5 | 42.7 | 65.9 | 46.2 |
| TwinsP-S (Chu et al., 2021a) | 44 | 245 | 46.8 | 69.3 | 51.8 | 42.6 | 66.3 | 46.0 |
| Twins-S (Chu et al., 2021a) | 44 | 228 | 46.8 | 69.2 | 51.2 | 42.6 | 66.3 | 45.8 |
| Swin-T (Liu et al., 2021) | 48 | 264 | 46.0 | 68.2 | 50.2 | 41.6 | 65.1 | 44.8 |
| Focal-T (Yang et al., 2021) | 49 | 291 | 47.2 | 69.4 | 51.9 | 42.7 | 66.5 | 45.9 |
| PVTv2-B2 (Wang et al., 2022b) | 45 | 309 | 47.8 | 69.7 | 52.6 | 43.1 | 66.8 | 46.7 |
| XciT-S12/8 (Ali et al., 2021) | 43 | 550 | 47.0 | 68.9 | 51.7 | 42.3 | 66.0 | 45.4 |
| DaViT-T (Ding et al., 2022) | 48 | 363 | 46.5 | 68.1 | 49.6 | 32.3 | 50.6 | 59.9 |
| DAT-T (Xia et al., 2022) | 48 | 272 | 47.1 | 69.2 | 51.6 | 42.4 | 66.1 | 45.5 |
| MViTv2-T (Li et al., 2022) | 44 | 279 | 48.2 | **70.9** | 53.3 | 43.8 | 67.9 | **47.2** |
| DWAViT-T[cos] (Ours) | 42 | 255 | 48.4 | 70.4 | 53.1 | 43.5 | 67.7 | 47.1 |
| DWAViT-T[quad] (Ours) | 42 | 255 | **48.8** | 70.7 | **53.6** | **43.8** | **68.1** | 47.1 |
| Res101 (He et al., 2016) | 63 | 336 | 42.8 | 63.2 | 47.1 | 38.5 | 60.1 | 41.3 |
| ConvNeXt-S (Liu et al., 2022b) | 70 | 348 | 47.9 | 70.0 | 52.7 | 42.9 | 66.9 | 46.2 |
| PVT-M (Wang et al., 2021) | 64 | 302 | 44.2 | 66.0 | 48.2 | 40.5 | 63.1 | 43.5 |
| ViL-M (Zhang et al., 2021) | 60 | 261 | 44.6 | 66.3 | 48.5 | 40.7 | 63.8 | 43.7 |
| TwinsP-B (Chu et al., 2021a) | 64 | 302 | 47.9 | 70.1 | 52.5 | 43.2 | 67.2 | 46.3 |
| Twins-B (Chu et al., 2021a) | 76 | 340 | 48.0 | 69.5 | 52.7 | 43.0 | 66.8 | 46.6 |
| Swin-S (Liu et al., 2021) | 69 | 354 | 48.5 | 70.2 | 53.5 | 43.3 | 67.3 | 46.6 |
| Focal-S (Yang et al., 2021) | 71 | 401 | 48.8 | 70.5 | 53.6 | 43.8 | 67.7 | 47.2 |
| PVTv2-B3 (Wang et al., 2022b) | 65 | 397 | 48.4 | 69.8 | 53.3 | 43.2 | 66.9 | 46.7 |
| XCiT-M24/8 (Ali et al., 2021) | 99 | 1448 | 48.5 | 70.3 | 53.4 | 43.7 | 67.5 | 46.9 |
| DAT-S (Xia et al., 2022) | 69 | 378 | 49.0 | 70.0 | 53.3 | 43.6 | 67.4 | 47.0 |
| DWAViT-S[cos] (Ours) | 64 | 338 | 48.4 | 70.0 | 53.3 | 43.6 | 67.4 | 47.0 |
| DWAViT-S[quad] (Ours) | 64 | 338 | **49.1** | **70.8** | **53.5** | **44.0** | **68.2** | **47.4** |
| X101-64 (Xie et al., 2017) | 101 | 493 | 44.4 | 64.9 | 48.8 | 39.7 | 61.9 | 42.6 |
| PVT-L (Wang et al., 2021) | 81 | 364 | 44.5 | 66.0 | 48.3 | 40.7 | 63.4 | 43.7 |
| Swin-B (Liu et al., 2021) | 107 | 496 | 48.5 | 69.8 | 53.2 | 43.4 | 66.8 | 46.9 |
| DWAViT-B[cos] (Ours) | 97 | 462 | **49.8** | **71.2** | **54.8** | **44.5** | 68.6 | **47.8** |
| DWAViT-B[quad] (Ours) | 97 | 462 | 49.4 | 71.1 | 54.6 | 44.4 | **68.6** | **47.8** |

Table 5 show the performance of of our our model and the baseline models with Cascade Mask R-CNN (Cai & Vasconcelos, 2018) framework on object detection and instance segmentation. Swin Transformer (Liu et al., 2021) and DAT (Xia et al., 2022) are the major baselines for the exact comparison. The experimental results show that our DWAViT outperforms baseline methods. DWAViT-T can achieves 52.2% on $AP^b$ and 45.1% on $AP^m$, and DWAViT-S can achieves 52.5% on $AP^b$ and 45.6% on $AP^m$. The DWAViT-S with quadratic self-attention can achieve 45.6% and 49.9% on $AP^m$ and $AP_{75}^m$, respectively, which is 0.1% and 0.3% higher than that of DAT-S (Xia et al., 2022).

## 4.3 ADE20K Semantic Segmentation

In this section, we further investigate the performance of our proposed model on semantic segmentation task. The Upernet (Xiao et al., 2018) framework is adopted. Our model and the baseline methods are evaluated on benchmark ADE20K (Zhou et al., 2017). For fair comparison, we follow the training procedure from previous works (Ding et al., 2022; Dong et al., 2022) and perform the experiment with MMSegmentation toolbox (Contributors, 2020). The image is resized to $512 \times 512$ and train the model with 160K iterations. the mIoU is adopted as the metric. The results of experiment are illustrated in Table 6 and Table 14 (see Appendix). SWin Transformer (Liu et al., 2021), Focal Transformer (Yang et al., 2021), XciT (Ali et al., 2021), DaViT (Xia et al., 2022) and DAT (Xia et al., 2022) are the baselines for exact comparison. Those methods are also implemented with MMSegmentation toolbox (Contributors, 2020). Besides, Upernet (Xiao et al., 2018) is adopted as the framework and models pre-trained on ImageNet-1K only are used as the feature

Table 5: Object detection and instance segmentation performance the our model and the baseline models with Cascade Mask R-CNN framework. The model is trained with 3x scheme. The FLOPs are measured at resolution $800 \times 1280$. [cos] and [quad] denote cosine and quadratic function, respectively.

| Backbone | #Param(M) | #FLOPs(G) | $AP^b$ | $AP^b_{50}$ | $AP^b_{75}$ | $AP^m$ | $AP^m_{50}$ | $AP^m_{75}$ |
|---|---|---|---|---|---|---|---|---|
| Res50 (He et al., 2016) | 82 | 739 | 46.3 | 64.3 | 50.5 | 40.1 | 61.7 | 43.4 |
| Swin-T (Liu et al., 2021) | 86 | 745 | 50.5 | 69.3 | 54.9 | 43.7 | 66.6 | 47.1 |
| DAT-T (Xia et al., 2022) | 86 | 750 | 51.3 | 70.1 | 55.8 | 44.5 | 67.5 | 48.1 |
| DWAViT-T[cos] (Ours) | 80 | 734 | **52.2** | **71.0** | **57.0** | **45.1** | 68.3 | **49.0** |
| DWAViT-T[quad] (Ours) | 80 | 734 | 51.0 | 70.6 | 56.7 | 44.9 | **68.5** | 48.8 |
| X101-32 (Xie et al., 2017) | 101 | 819 | 48.1 | 66.5 | 52.4 | 41.6 | 63.9 | 45.2 |
| Swin-S (Liu et al., 2021) | 107 | 838 | 51.8 | 70.4 | 56.3 | 44.7 | 67.9 | 48.5 |
| DAT-S (Xia et al., 2022) | 107 | 807 | **52.7** | **71.7** | **57.2** | 45.5 | **69.1** | 49.3 |
| DWAViT-S[cos] (Ours) | 102 | 817 | 52.5 | 71.3 | 57.0 | **45.6** | 68.9 | 49.6 |
| DWAViT-S[quad] (Ours) | 102 | 817 | 52.5 | 71.4 | **57.2** | **45.6** | 68.9 | **49.9** |
| X101-64 (Xie et al., 2017) | 140 | 972 | 48.3 | 66.4 | 52.3 | 41.7 | 64.0 | 45.1 |
| Swin-B (Liu et al., 2021) | 145 | 982 | 51.9 | 70.9 | 56.5 | 45.0 | 68.4 | 48.7 |
| DWAViT-B[cos] (Ours) | 134 | 940 | 52.5 | 71.3 | 57.0 | 45.6 | 69.0 | **49.6** |
| DWAViT-B[quad] (Ours) | 134 | 940 | **52.8** | **71.6** | **57.3** | **45.7** | **69.1** | 49.5 |

Table 6: The semantic segmentation performance of DWAViT-T and baselines on ADE20k. The FLOPs are calculated with resolution $512 \times 2048$. [cos] and [quad] denote cosine and quadratic function, respectively.

| Backbone | #Param(M) | #FLOPs(G) | mIoU |
|---|---|---|---|
| Swin-T (Liu et al., 2021) | 59 | 945 | 44.5 |
| Focal-T (Yang et al., 2021) | 62 | 998 | 45.8 |
| XciT-S12/16 (Ali et al., 2021) | 54 | 966 | 45.9 |
| XciT-S12/8 (Ali et al., 2021) | 53 | 1237 | 46.6 |
| DaViT-T (Ding et al., 2022) | 60 | 940 | 46.3 |
| DAT-T (Xia et al., 2022) | 60 | 957 | 45.5 |
| DWAViT-T[cos] (Ours) | 52 | 930 | **47.5** |
| DWAViT-T[quad] (Ours) | 52 | 930 | 45.4 |

extractor for theu baselines and our method. The results show that our DWAViT with cosine self-attention function can outperform the baselines. Besides, the performance of our model with cosine function is also much better than that of our model with quadratic function. The mIoU of DWAViT-T can reach 47.5% , which outperforms other baseline methods like DAT-T (Xia et al., 2022), DaViT-T (Ding et al., 2022) and XciT-S (Ali et al., 2021). The mIoU of DWAViT-S and DWAViT-B can reach 49.3% and 49.5%, respectively, which can outperform other baseline models like DAT (Xia et al., 2022) and DaViT (Ding et al., 2022).

## 4.4 Runtime Analysis

In this section we quantitatively evaluate the actual runtime of our model during the training on image classification, object detection and semantic segmentation tasks. Specifically, for image classification task, we fix the batch to 100 and compare the runtime of our model with that of Swin Transformer (Liu et al., 2021). For object detection and semantic segmentation, we adopt the Mask R-CNN and Upernet as the framework and set the batch size to 2. All the experiments are implemented on a single A100 GPU and we report the average time per iteration. Table 11 illustrates the runtime of our model and the Swin Transformer (Liu et al., 2021) on image classification task. The results suggest that our model would be more time-consuming than that of Swin Transformer (Liu et al., 2021) due to the different form of self-attention function and local window mechanism. Specifically, compared to Swin-T (Liu et al., 2021), our DWAViT-T(quad) can achieve 1.6% higher accuracy in image classification with doubled computational cost. Our DWAViT-B(cos) can achieve an improvement of 0.5% in accuracy for image classification task but it takes twice as much time

Table 7: Object detection and instance segmentation performance the our DWAViT with angular self-attention and scaled dot-product self-attention in Mask R-CNN framework. The model is trained with 3x scheme. The FLOPs are measured at resolution $800 \times 1280$. The [dot product], [cos] and [quad] denote scaled dot-product, cosine and quadratic function, respectively.

| Backbone | #Param(M) | #FLOPs(G) | $AP^b$ | $AP^b_{50}$ | $AP^b_{75}$ | $AP^m$ | $AP^m_{50}$ | $AP^m_{75}$ |
|---|---|---|---|---|---|---|---|---|
| DWAViT-T[dot product] | 42 | 255 | 48.4 | 70.3 | 53.2 | 43.5 | 67.5 | **47.1** |
| DWAViT-T[cos] | 42 | 255 | 48.4 | 70.4 | 53.1 | 43.5 | 67.7 | **47.1** |
| DWAViT-T[quad] | 42 | 255 | **48.8** | **70.7** | **53.6** | **43.8** | **68.1** | **47.1** |

Table 8: The performance of our proposed DWAViT with angular self-attention and scaled dot-product self-attention on the ImageNet-1K classification. The resolution of the image is $224 \times 224$.

| Model | Param(M) | FLOPs(G) | Top-1 Acc |
|---|---|---|---|
| DWAViT-T[dot product] | 22.7 | 4.2 | 82.5 |
| DWAViT-T[cos] | 22.7 | 4.2 | 82.7 |
| DWAViT-T[quad] | 22.7 | 4.2 | **82.8** |

Table 9: The performance of our proposed DWAViT with angular self-attention and scaled dot-product self-attention on the ADE20k semantic segmentation. FLOPs are calculated with resolution $512 \times 2048$.

| Backbone | Param(M) | FLOPs(G) | mIoU |
|---|---|---|---|
| DWAViT-T[dot product] | 52 | 930 | 44.7 |
| DWAViT-T[cos] | 52 | 930 | **47.5** |
| DWAViT-T[quad] | 52 | 930 | 45.4 |

Table 10: The performance of DeiT-S and CSwin-T with angular self-attention and scaled dot-product self-attention on the ImageNet-1K classification. The resolution of the image is $224 \times 224$. The dot product, cos and quad in the square bracket denote scaled dot-product, cosine and quadratic function, respectively.

| Model | Param(M) | FLOPs(G) | Top-1 Acc |
|---|---|---|---|
| DeiT-S (Touvron et al., 2021a) [dot product] | 22 | 4.6 | 79.8 |
| DeiT-S (Touvron et al., 2021a) [cos] | 22 | 4.6 | 80.0 |
| DeiT-S (Touvron et al., 2021a) [quad] | 22 | 4.6 | **80.2** |
| CSwin-T (Dong et al., 2022)[exp] | 23 | 4.3 | 82.7 |
| CSwin-T (Dong et al., 2022)[cos] | 23 | 4.3 | 82.9 |
| CSwin-T (Dong et al., 2022)[quad] | 23 | 4.3 | **83.0** |

Table 11: The actual runtime of Swin Transformer and our DWAViT during the training on ImageNet-1K. We implement the testing on a single A100 GPU and the batch size is fixed to 100 for all the models. The average time per iteration is reported.

| Model | #Param(M) | #FLOPs(G) | time_per_iteration(s) |
|---|---|---|---|
| Swin-T (Liu et al., 2021) | 28.3 | 4.5 | 0.15 |
| DWAViT-T[cos](Ours) | 22.7 | 4.2 | 0.29 |
| DWAViT-T[quad] (Ours) | 22.7 | 4.2 | 0.31 |
| Swin-S (Liu et al., 2021) | 49.6 | 8.7 | 0.23 |
| DWAViT-S[cos](Ours) | 44.6 | 8.2 | 0.43 |
| DWAViT-S[quad] (Ours) | 44.6 | 8.2 | 0.48 |
| Swin-B (Liu et al., 2021) | 87.8 | 15.4 | 0.30 |
| DWAViT-B[cos](Ours) | 77.4 | 14.3 | 0.60 |
| DWAViT-B[quad] (Ours) | 77.4 | 14.3 | 0.64 |

as that of Swin-B Liu et al. (2021). Basically, the time taken by our model is almost twice that of Swin Transformer (Liu et al., 2021). However, the actual runtime does not blow up when the size of our model scales up. The runtime of our models on object detection and semantic segmentation is illustrated in Table 12 and Table 13, respectively.

## 4.5 Ablation Study

In the ablation study we aim to compare the performance of our model with traditional scaled dot-product self-attention and our proposed angular self-attention. In the experiment we replace the angular self-attention with traditional scaled dot-product self-attention and evaluate the performance on ImageNet-1K classification,

Table 12: The actual runtime of our DWAViT during the training on COCO for object detection task. Mask R-CNN is adopted as the framework. We implement the testing on a single A100 GPU and the batch size is fixed to 2 for all the models. The average time per iteration is reported.

| Backbone | #Param(M) | #FLOPs(G) | time_per_iteration(s) |
|---|---|---|---|
| DWAViT-T[cos] | 42 | 255 | 0.24 |
| DWAViT-T[quad] | 42 | 255 | 0.25 |
| DWAViT-S[cos] | 44.6 | 8.2 | 0.29 |
| DWAViT-S[quad] | 44.6 | 8.2 | 0.30 |
| DWAViT-B[cos] | 77.4 | 14.3 | 0.57 |
| DWAViT-B[quad] | 77.4 | 14.3 | 0.57 |

Table 13: The actual runtime of our DWAViT during the training on ADE20K for object detection task. Upernet is adopted as the framework. We implement the testing on a single A100 GPU and the batch size is fixed to 2 for all the models. The average time per iteration is reported.

| Backbone | #Param(M) | #FLOPs(G) | time_per_iteration(s) |
|---|---|---|---|
| DWAViT-T[cos] | 42 | 255 | 0.19 |
| DWAViT-T[quad] | 42 | 255 | 0.20 |
| DWAViT-S[cos] | 44.6 | 8.2 | 0.22 |
| DWAViT-S[quad] | 44.6 | 8.2 | 0.23 |
| DWAViT-B[cos] | 77.4 | 14.3 | 0.56 |
| DWAViT-B[quad] | 77.4 | 14.3 | 0.56 |

COCO object detection and ADE20K semantic segmentation. We adopt the DWAViT-T as the backbone in three tasks. In object detection Mask R-CNN is adopted as the framework and the model is trained with 36 epochs. In semantic segmentation Upernet is adopted as the framework and the model is trained with 160K iteration. The results of the image classification, object detection and semantic segmentation are illustrated in Table 4.3, Table 7 and Table 9, respectively. When our model adopt the traditional scaled dot-product self-attention, the top-1 accuracy would be 0.2% and 0.3% lower than that of cosine function and quadratic function. On other two tasks the performance of our model with scaled dot-product self-attention is also lower than that of our model with the proposed angular self-attention. On some tasks the the gap of the performance is more obvious. For instance, on semantic segmentation our model of scaled dot-product self-attention only achieve 44.7% of mIoU, approximately 3% lower than that of our model with cosine self-attention. The experimental results suggest our angular self-attention can model the relationship of the tokens successfully and the angular self-attention is a powerful alternative to the traditional scaled dot-product self-attention.

We further investigate the performance of our angular self-attention by replacing the scaled dot-product self-attention with our angular self-attention in other vision transformer models. Table 10 shows the performance of DeiT-S (Touvron et al., 2021a) and CSWin-T (Dong et al., 2022) on ImageNet-1K image classification with scaled dot-product self-attention and our proposed angular self-attention. The experimental results suggest that when the existing vision transformer models are equipped with our angular self-attention, the performance on the image classification can be improved. It further validates that our angular self-attention is a competitive alternative for the existing scaled dot-product self-attention.

## 5 Conclusions

In this paper, we propose the dual-window mechanism and angular self-attention. The dual-window mechanism divide the feature maps into even/odd number of local window in each stage alternatively for the information exchange of the local windows. In angular self-attention, the traditional scaled dot-product operation is replaced by our proposed quadratic and cosine functions. The proposed angular function can also model the relationship of tokens in the long range. Based on dual-window mechanism and angular self-attention, we propose a new vision transformer backbone called dual-windowed angular vision transformer. Extensive

experiments show that our backbone can achieve competitive performance on the tasks such as image classification, object detection and semantic segmentation. However, the computational cost of our model is higher than that of baseline models due to the new formulation of the self-attention.

**Broader Impact.** This work proposed a new architecture of vision transformer called DWAViT featured by angular self-attention and dual local window mechanism. Our model is proven to achieve competitive performance in downstream tasks such as object detection and semantic segmentation and has the enormous potential to be used in various practical scenarios. In particular, object detection is one of the most promising applications of vision transformers in the real world and it is often used in systems which require extensive interaction with the surrounding environment visually. For instance, autonomous vehicles require a large number of object detectors to identify the pedestrians and other vehicles nearby and the safety and the trustworthiness of vision transformers are critical in this area. Though our proposed model can achieve promising results on object detection and other tasks, some critical issues such as adversarial robustness and trustworthiness are quite under-explored and further investigation is necessarily required.

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

# A    Appendix

The proof of Proposition 1 is provided below.

**Proof 1** *Assume $\theta_0 \leq \theta_1 \leq ... \leq \theta_{J-1}$ for angles between the target query and all the keys. $J$ is the total number of the tokens. $V$ is the value vector. According to our angular self-attention, the embedding of target tokens can be expressed as:*

$$
\begin{aligned}
\mathcal{O} &= \frac{1}{C} \sum_j \exp(s(\theta_j)/\tau)V_j \\
&= \frac{1}{C} \exp(\frac{s(\theta_0)}{\tau})(V_0 + \exp(\frac{s(\theta_1) - s(\theta_0)}{\tau})V_1 + ... \\
&\quad + \exp(\frac{s(\theta_{J-1}) - s(\theta_0)}{\tau})V_{J-1}) \\
&= \frac{1}{C} \exp(\frac{s(\theta_0)}{\tau})(V_0 + \omega_1 V_1 + ... + \omega_{J-1} V_{J-1}), \\
&\propto V_0 + \omega_1 V_1 + \cdots + \omega_{J-1} V_{J-1},
\end{aligned}
\tag{8}
$$

*where $\omega_j = \exp(\frac{s(\theta_j) - s(\theta_0)}{\tau})$ and $C$ is normalization coefficient. Since $\theta_j > \theta_0$ and $s(\theta_j) < s(\theta_0)$, $\omega_j \in (0, 1)$.*

Table 14: The semantic segmentation performance of DWAViT-S, DWAViT-B and baselines on ADE20k. The FLOPs are calculated with resolution $512 \times 2048$. [cos] and [quad] denote cosine and quadratic function, respectively.

| Backbone | #Param(M) | #FLOPs(G) | mIoU |
|---|---|---|---|
| ResNet-101 (He et al., 2016) | 86 | 1029 | 44.9 |
| XCiT-S24/16 (Ali et al., 2021) | 76 | 1053 | 46.9 |
| TwinsP-B (Chu et al., 2021a) | 74 | 977 | 47.1 |
| XCiT-M24/16 (Ali et al., 2021) | 112 | 1213 | 47.6 |
| Swin-S (Liu et al., 2021) | 81 | 1038 | 47.6 |
| Twins-B (Chu et al., 2021a) | 89 | 1020 | 47.7 |
| Focal-S (Yang et al., 2021) | 85 | 1130 | 48.0 |
| DaViT-T (Ding et al., 2022) | 81 | 1030 | 48.8 |
| DAT-T (Xia et al., 2022) | 83 | 1079 | 48.3 |
| DWAViT-S[cos] (Ours) | 75 | 1015 | **49.3** |
| DWAViT-S[quad] (Ours) | 75 | 1015 | 47.8 |
| XCiT-M24/8 (Ali et al., 2021) | 110 | 2161 | 48.4 |
| Swin-B (Liu et al., 2021) | 121 | 1841 | 48.1 |
| Focal-B (Yang et al., 2021) | 126 | 1354 | 49.0 |
| DaViT-B (Ding et al., 2022) | 121 | 1175 | 49.4 |
| DAT-B (Xia et al., 2022) | 121 | 1212 | 49.4 |
| DWAViT-B[cos] (Ours) | 108 | 1143 | **49.5** |
| DWAViT-B[quad] (Ours) | 108 | 1143 | 47.9 |

