# OpenReview forum: "Dual-windowed Vision Transformer with Angular Self-Attention"
_TMLR — Rejected by TMLR_

### Review · Reviewer_s8F4 · 2023-08-19

**Summary Of Contributions:**

The paper explores variants of vision transformers based on applying a normalization prior to the self-attention mechanism (such that only the angle between vectors is taken into account) and mechanism to apply self-attention among neighboring tokens that is effective without compromising the computational cost. Results show that the proposed final architecture is competitive with state-of-the-art accuracy in several computer vision tasks.

**Audience:**

Yes

**Claims And Evidence:**

Yes

**Requested Changes:**

Most of my suggestions are about improving the text to state more clearly the achievements of the paper and help the reader understand the paper.

An important request that needs to be fulfilled is to provide the code to reproduce the experiments.

**Strengths And Weaknesses:**

The paper is easy to follow and the claims seem to be accurate given the evidence presented. The following are several suggestions that I hope help the authors:

- The techniques explored in this paper are well described and easy to follow. The dual window mechanism is easy to understand the intuitions behind it and the motivation why this was introduced. Yet, the normalization of the self-attentions seems quite an arbitrary choice to explore. I would suggest trying to motive a bit the reasons why the authors decided investigating this (eg. what were the expectations before and after the experiments).
- Theoretical analysis: This sections comes quite abruptly, it would help to explain what is the intent of the proposition before introducing it. This section is hard to follow without providing more context.
- Please provide the code to reproduce all the experiments in the paper.
- State more clearly (ie. providing quantities) what is the computational cost gain and accuracy gain.

There are also several typos and text that needs some refinement:
- In abstract "we propose two solutions": no problem was discussed before, so talking about "solutions" is confusing.
- In abstract "We evaluate...": it is mentioned the evaluation but not the takeaway of those evaluations which is what is relevant to learn in the abstract.
- List of contributions (top of page 3), first point: "We propose the dual window mechanism to reduce the computation cost..." indicate by how much the computational cost is reduced.
- Section 3.1 "cycle shift." and "following layer," it seems "," and "." have been swapped.
- Equation 6 is missing a parenthesis

---

> ### Author Response · Authors · 2023-09-16
> **Response one**
>
> **Q1: The techniques explored in this paper are well described and easy to follow. The dual window mechanism is easy to understand the intuitions behind it and the motivation why this was introduced. Yet, the normalization of self-attention seems quite an arbitrary choice to explore. I would suggest trying to motivate a bit the reasons why the authors decided investigating this (eg. what were the expectations before and after the experiments).**
>
> **R1:** Thank you very much for your constructive comments! Inspired by previous work [1,2], we notice that scaled dot-product function is the only choice to model the relationship of tokens. Thus, we propose two functions: quadratic and cosine functions to achieve the same objective. In scaled dot-product function, the attention weight would be dependent on the L2 norm of Q and K as well as their angles θ. In our method, we first normalize the Q and K so that they are distributed on the surface of the unit sphere. Through our training, the angles of tokens would be adjusted to model the relationships of tokens. With the new functions the model is supposed to achieve better performance and both our experiment and ablation study have validated this. We have revised the Introduction of our paper accordingly.
>
>
> [1] Zhao, Hengshuang, Jiaya Jia, and Vladlen Koltun. "Exploring self-attention for image recognition." Proceedings of the IEEE/CVF conference on computer vision and pattern recognition. 2020.
>
> [2] Wang, Xiaolong, et al. "Non-local neural networks." Proceedings of the IEEE conference on computer vision and pattern recognition. 2018.
>
> **Q2: Theoretical analysis: This section comes quite abruptly, it would help to explain what is the intent of the proposition before introducing it. This section is hard to follow without providing more context.**
>
> **R2:** Thank you for the valuable suggestion! In our work we proposed quadratic and cosine functions to model the relationships of tokens. Compared to the scale dot-product function, the advantage of our method is that the relationship of tokens is only dependent on the angles between tokens.  To better understand the angular self-attention, it is essential to investigate the relationship of tokens in our method .Thus , we provided the Proposition to analyze this problem. In the revised paper, we have added a paragraph before the Proposition to describe the intention of theoretical analysis.
>
> **Q3: Please provide the code to reproduce all the experiments in the paper.**
>
> **R3:**  We have provided the code in the anonymous github repository. The link is https://anonymous.4open.science/r/vision-transformer-C8FF/. Once the paper is accepted, we will reorganize the code and release it publicly.

---

> > ### Author Response · Authors · 2023-09-16
> > **Response two**
> >
> > **Q4: State more clearly (ie. providing quantities) what is the computational cost gain and accuracy gain.**
> >
> > **R4:**  Thanks for the suggestion! The table below shows the runtime of our model and the Swin Transformer [1] in image classification task. The experiment is implemented on a single A100 GPU. For fair evaluation, we fix the batch size to 100 for all the models. We report the average time per iteration in the table. The results show that our model would take the time twice as much as that of Swin Transformer [1] due to the different self-attention function and the different local window mechansim. However, with the reasonable extra time cost, our model (DWAViT-T) can achieve 82.8% in the image classification task, higher than that of Swin-T (81.2%). More results can be found in Tables 12 and 13 of the revised paper.
> >
> > [1] Ze Liu, Yutong Lin, Yue Cao, Han Hu, Yixuan Wei, Zheng Zhang, Stephen Lin, and Baining Guo. Swin transformer: Hierarchical vision transformer using shifted windows. In Proceedings of the IEEE/CVF international conference on computer vision, pp. 10012–10022, 2021.
> >
> > |  Model                |   #Param(M)  |  #FLOPs(G)   | time_per_iteration(s)   |
> > |  :----  | :----: | :----:  | :----: |
> > | Swin-T                  |      28.3         |        4.5          |            0.15                  |
> > | DWAViT-T[cos]     |      22.7         |        4.2          |            0.29                  |
> > | DWAViT-T[quad]   |      22.7         |        4.2          |            0.31                  |
> > | Swin-S                  |      49.6         |        8,7          |            0.23                 |
> > | DWAViT-S[cos]     |      44.6         |        8.2          |            0.43                  |
> > | DWAViT-S[quad]   |      44.6         |        8.2          |            0.38                  |
> > | Swin-B                  |      87.8         |        15.4         |           0.30                  |
> > | DWAViT-B[cos]     |      77.4         |        14.3          |            0.60                  |
> > | DWAViT-B[quad]   |      77.4         |        14.3          |            0.64                  |
> >
> > **Q5: There are also several typos and text that needs some refinement.**
> >
> > **R5:** : Thank you for your suggestion. We have modified the typos and added some clarification in our manuscript.

---

> > > ### Comment · Reviewer_s8F4 · 2023-09-20
> > >
> > > Thank you very much for your answers. I have read the rebuttal and I would like to point out a couple of major issues:
> > >
> > > - The code is incomplete: the readme is empty and I could not find the code to run the experiments for the different datasets in the paper. It is important that the code to reproduce the main experiments in the paper is provided.
> > >
> > > - The claims made in abstract, end of introduction and conclusions are vague and do not seem to be accurate. For example, at the end of introduction, one of the contributions is "We propose the dual window mechanism to reduce the computation cost in the self-attention
> > > calculation .", but when looking at the tables with computational cost provided in the rebuttal, the computational cost of the proposed method is higher than competitors. Please rewrite them to be strictly accurate with the evidence provided in the results. I suggest adding quantitative statements (eg. the accuracy improved x% in this dataset with a reduction of y% of the computational cost)

---

> > > > ### Author Response · Authors · 2023-09-22
> > > > **Response one**
> > > >
> > > > **Q1: The code is incomplete: the readme is empty and I could not find the code to run the experiments for the different datasets in the paper. It is important that the code to reproduce the main experiments in the paper is provided.**
> > > >
> > > > **R1:** Thanks for your suggestion. I have added the readme file on the instructions for the reprodution of our experiments in the paper. I also add the source code on all the datasets. Please download the readme file for review. If you have any further question, feel free to let me know.
> > > >
> > > > **Q2: The claims made in abstract, end of introduction and conclusions are vague and do not seem to be accurate. For example, at the end of introduction, one of the contributions is "We propose the dual window mechanism to reduce the computation cost in the self-attention calculation .", but when looking at the tables with computational cost provided in the rebuttal, the computational cost of the proposed method is higher than competitors. Please rewrite them to be strictly accurate with the evidence provided in the results. I suggest adding quantitative statements (eg. the accuracy improved x% in this dataset with a reduction of y% of the computational cost)**
> > > >
> > > > **R2:** Thanks for your comment. When I mention the dual window can reduce the computation cost, it is referred to the reduction of computation cost compared to that without any local window mechanism. It is not necessarily the reduction of computation cost for our dual window compared to other local window techniques. As oberserved from our experimental results, the computation cost of our dual window is larger than that of local window proposed by Swin Transformer. I have revised the the corresponding paragraphs to clarify the misunderstanding. Furthermore, I aslo add the quantitative statements at Sec. 4.4 in our manuscript.

---

> > > > > ### Comment · Reviewer_s8F4 · 2023-10-20
> > > > > **Final comment**
> > > > >
> > > > > After reading the new version of abstract, end of introduction and conclusions, I still think they are vague and ambiguous. The statements there do not reflect the actual conclusions taken from the results. For example, the abstract mentions that "We also validate the effectiveness...", in intro " the dual window mechanism to reduce the computation cost in the self-attention
> > > > > calculation compared to that without any local window technique. ", in conclusions " our backbone can achieve competitive performance ...". So, authors have not addressed my second concern.

---

> ### Author Response · Authors · 2023-10-20
> **Response**
>
> **Q1: After reading the new version of abstract, end of introduction and conclusions, I still think they are vague and ambiguous. The statements there do not reflect the actual conclusions taken from the results. For example, the abstract mentions that "We also validate the effectiveness...", in intro " the dual window mechanism to reduce the computational cost in the self-attention calculation compared to that without any local window technique. ", in conclusion " our backbone can achieve competitive performance ...". So, authors have not addressed my second concern.**
>
> **R1:** Thank you very much for your constructive comments. We have revised the relevant statements in abstract, introduction and conclusion sections that may cause confusion or misunderstanding. Following your suggestions, some of the revisions are as follows:
>
> Abstract:
>
> “We also validate the effectiveness of our angular self-attention by investigating the performance of vision transformers with the scaled dot-product operation replaced by our angular function on several tasks.”  is changed to “Our experimental results also suggest that though our model can achieve promising performance on the tasks, the computational cost of our model is higher than that of the baseline models (i.e., Swin Transformer) due to the new formulation of the self-attention”.
>
>
> Introduction:
>
> “We propose the dual window mechanism to reduce the computational cost in the self-attention calculation compared to that without any local window technique” is changed to “We propose the dual window mechanism to split the global feature map into a couple of smaller localized feature maps.“
>
> “The dual window mechanism can substantially reduce the computational cost in the backbone compared to that without any local window technique and also preserve the ability to model long-range relationship between tokens” is changed to “The dual window mechanism can preserve the ability to model long-range relationship between
> tokens. However, due to the new formulation of the self-attention. the computational cost of our model is higher than that of baseline models with a similar size.”
>
>
> Conclusion:
>
> “Extensive experiments show that our backbone can achieve competitive performance on the tasks such image classification, object detection and semantic segmentation against other strong baselines.” is changed to “Extensive experiments show that our backbone can achieve competitive performance on the tasks such as image classification, object detection and semantic segmentation. However, the computational cost of our model is higher than that of baseline models due to the new formulation of the self-attention.”

---

### Review · Reviewer_j1bi · 2023-08-30

**Summary Of Contributions:**

In this paper, the authors proposed dual-windowed ViT with angular attention (DWA-ViT). As evidenced in its name, the contribution is two-folded:

- The authors proposed to use alternate window sizes across adjacent blocks to achieve a similar effect of SwinTransformer;
- The authors change the similarity formulation in ViTs to one based on angular distance.

The effectiveness of DWAViT is demonstrated through evaluations on various computer vision benchmarks, including ImageNet-1K for image classification, COCO for object detection, and ADE20k for semantic segmentation. The paper also validates the proposed angular self-attention by comparing it to the traditional scaled dot-product operation on multiple tasks.

**Audience:**

Yes

**Claims And Evidence:**

Yes

**Requested Changes:**

Please respond to my questions in the "Weaknesses" section.

**Strengths And Weaknesses:**

Strengths:

- The paper aligns well with the interests of a specific audience (This is one of the acceptance criterion).

- The concept of dual windows is innovative. I am not aware of apparent precedent in existing literature (though other reviewers could correct me if I am wrong).

- The authors effectively demonstrate the superiority of their proposed designs (dual windows and angular self-attention) over existing approaches in certain contexts (This is one of the acceptance criterion).

- Rigorous experimentation covers a range of tasks (classification, detection, segmentation) and datasets, enhancing the paper's robustness.

Weaknesses:

- While the notion of normalizing features and using cosine similarity as a distance metric is common in face recognition, its application might not be novel enough for potential paper rejection according to TMLR's guidelines.

- The two proposed techniques (dual windows and angular self-attention) appear somewhat disjointed and self-contained. It would be valuable to explore potential synergies or connections between these innovations.

- The utilization of cosine similarity and quadratic distance as angular self-attention metrics seems to cater to distinct application scenarios. More insights into this distinction would enrich the understanding.

- The runtime of the proposed DWA-ViT is absent from the discussion. Including these latency figures would enhance the paper's practicality, even though it's assumed to be no slower than Swin.

---

> ### Author Response · Authors · 2023-09-16
> **Response one**
>
> **Q1: While the notion of normalizing features and using cosine similarity as a distance metric is common in face recognition, its application might not be novel enough for potential paper rejection according to TMLR's guidelines.**
>
> **R1:** Our proposed angular self-attention has fundamental differences with the idea of normalizing features and using cosine similarity in face recognition. First, we propose two alternative angular functions: quadratic and cosine functions. These two functions aim to model the long-range relationship of different tokens. In the traditional scaled dot-product self-attention in which the attention weight would be dependent on the L2 norm of Q and K as well as their angles θ. In our angular self-attention, we normalize the Q and K features so that they are distributed on the surface of the unit sphere. Then the attention weight obtained from our angular self-attention is solely dependent on the angle θ. Through our training the angles θ between different Q and K would be adjusted to model the relationship of different tokens and make the vision transformer achieve strong representative ability. Extensive experiments have validated the effectiveness of our angular self-attention and it achieves competitive performance in the downstream tasks compared to scaled dot-product self-attention. While in face recognition, only cosine function is adopted and it is mainly used as the distance metric for positive/negative pairs.
>
> Besides, the motivation of our angular self-attention comes from the previous work [1,2], which suggests that the relationship of tokens can be modeled by other forms of functions and scaled dot-product function is not the only choice. Inspired by this finding, we propose the two angular functions to model the relationship of tokens. Thus, we believe that it is not a simple application of the techniques used in face recognition in vision transformers.
>
>
> [1] Zhao, Hengshuang, Jiaya Jia, and Vladlen Koltun. "Exploring self-attention for image recognition." Proceedings of the IEEE/CVF conference on computer vision and pattern recognition. 2020.
>
> [2] Wang, Xiaolong, et al. "Non-local neural networks." Proceedings of the IEEE conference on computer vision and pattern recognition. 2018.
>
>
> **Q2:: The two proposed techniques (dual windows and angular self-attention) appear somewhat disjointed and self-contained. It would be valuable to explore potential synergies or connections between these innovations.**
>
> **R2:** Thanks for the insightful comments! There are strong connections between the two techniques. On one hand, as we show in the ablation study, the angular self-attention can replace the scaled dot-product self-attention in the vision transformers and it can achieve competitive performance. However, the large computational cost of the angular self-attention on the large feature maps hinders its further application in the downstream tasks such as object detection and semantic segmentation. The dual window mechanism can greatly relieve this problem by dividing the large feature maps into smaller ones. On the other hand, our empirical study suggests that our angular self-attention can achieve better performance with our dual window techniques than that with  previous local window techniques (i.e., Swin Transformer [1]). To wrap up, the two proposed techniques (dual windows and angular self-attention) have strong mutual connections and are two indispensable components in our DWAViT.
>
> [1] Ze Liu, Yutong Lin, Yue Cao, Han Hu, Yixuan Wei, Zheng Zhang, Stephen Lin, and Baining Guo. Swin transformer: Hierarchical vision transformer using shifted windows. In Proceedings of the IEEE/CVF international conference on computer vision, pp. 10012–10022, 2021.
>
> **Q3: The utilization of cosine similarity and quadratic distance as angular self-attention metrics seems to cater to distinct application scenarios. More insights into this distinction would enrich the understanding.**
>
> **R3:** The cosine similarity and quadratic distance has common mathematical properties. Both functions are descending when $\theta \in [0, \pi]$, which means that the tokes with larger angles would have less weaker relationships.  Specifically, when $\theta \in [0, \pi/2]$, $\cos\theta \approx 1 - \theta^2/2 \approx 1 - 4\theta^2/\pi^2 $, when $\theta \in (\pi/2,\pi]$, $1 - 4\theta^2/\pi^2  < \cos\theta < 0$ , which means that tokens with angles larger than $\pi/2$ have weaker relationships in quadratic function than that of in cos function. In most tasks like image classification and object detection, the performance of quadratic and cosine functions is comparable and the difference is very slight (<0.5%). However, in semantic segmentation, the performance of cosine function is better than that of quadratic function (>1.0%).

---

> > ### Author Response · Authors · 2023-09-16
> > **Response two**
> >
> > **Q4: The runtime of the proposed DWA-ViT is absent from the discussion. Including these latency figures would enhance the paper's practicality, even though it's assumed to be no slower than Swin.**
> >
> > **R4:** Thanks for the suggestion! The table below shows the runtime of our model and the Swin Transformer [1] in image classification task. The experiment is implemented on a single A100 GPU. For fair evaluation, we fix the batch size to 100 for all the models. We report the average time per iteration in the table. The results show that our model would take the time twice as much as that of Swin Transformer due to the different self-attention function and the different local window mechansim. But the actual runtime doesn’t blow up when the size of the model scales up. More results can be found in Tables 12 and Table 13 of the revised paper.
> >
> > [1] Ze Liu, Yutong Lin, Yue Cao, Han Hu, Yixuan Wei, Zheng Zhang, Stephen Lin, and Baining Guo. Swin transformer: Hierarchical vision transformer using shifted windows. In Proceedings of the IEEE/CVF international conference on computer vision, pp. 10012–10022, 2021.
> >
> >
> > |  Model                |   #Param(M)  |  #FLOPs(G)   | time_per_iteration(s)   |
> > |  :----  | :----: | :----:  | :----: |
> > | Swin-T                  |      28.3         |        4.5          |            0.15                  |
> > | DWAViT-T[cos]     |      22.7         |        4.2          |            0.29                  |
> > | DWAViT-T[quad]   |      22.7         |        4.2          |            0.31                  |
> > | Swin-S                  |      49.6         |        8,7          |            0.23                 |
> > | DWAViT-S[cos]     |      44.6         |        8.2          |            0.43                  |
> > | DWAViT-S[quad]   |      44.6         |        8.2          |            0.38                  |
> > | Swin-B                  |      87.8         |        15.4         |           0.30                  |
> > | DWAViT-B[cos]     |      77.4         |        14.3          |            0.60                  |
> > | DWAViT-B[quad]   |      77.4         |        14.3          |            0.64                  |

---

### Review · Reviewer_b7iX · 2023-09-04

**Summary Of Contributions:**

The manuscript presents a transformer like architecture for visual data. It proposes:
1. Windowed self attention where information exchange across windows is achieved by using different window sizes in alternating blocks.
2. Angular attention, an alternative to dot product attention which computes a function of the angle, $\theta$, between key and query vectors.

Two variations of angular attention are proposed, `cos` and `quadratic`. `cos` computes $cos\left( \theta \right)$ and `quadratic` works on $\theta^2$.

The overall architecture also down-samples the feature grid after every stage.

The architecture is evaluated on ImageNet-1k classification, MSCOCO object detection and instance segmentation, and ADE-20k semantic segmentation.

**Audience:**

Yes

**Broader Impact Concerns:**

Computer vision, especially object detection, is an ethically sensitive topic. A broader impact statement is not present and I strongly encourage the authors to add one.

**Claims And Evidence:**

No

**Requested Changes:**

### Citations [critical]
In page 3, the manuscript identifies Dosovitskiy et al. 2021 as the first introduction of self-attention to computer vision. The method that first introduced self attention to computer vision is https://arxiv.org/abs/1802.05751 or https://arxiv.org/pdf/1711.07971.pdf; and transformer inspired architectures have been since applied to classification, for example [here](https://proceedings.neurips.cc/paper_files/paper/2019/file/3416a75f4cea9109507cacd8e2f2aefc-Paper.pdf). The idea of using image patches as tokens was also implemented in https://arxiv.org/abs/1911.03584.

Several alternatives to dot-product attention were explored in ["Exploring Self-attention for Image Recognition"](https://openaccess.thecvf.com/content_CVPR_2020/papers/Zhao_Exploring_Self-Attention_for_Image_Recognition_CVPR_2020_paper.pdf). The ["Non local networks"](https://arxiv.org/pdf/1711.07971.pdf) paper also discusses some alternatives to dot-product attention. These and similar works that appeared prior to ViT need to be cited.

### Analysis
The analysis quantifies performance when controlling for #params and #flops. Please also quote the actual runtime in training steps per second for a fixed hardware setup (#gpus) used across all experiments for atleast some subset of runs. In many cases, although the #flops are fixed, the actual runtime can blow up because of subtle differences in the formulation. [critical]

The analysis mainly compares against several baselines but it is not clear which baselines are actually an exact comparison. Exact comparison = same prediction head (mask-rcnn/cascaded mask-rcnn/upernet) + same training regime + same pre-training and fine-tuning datasets. Please clarify in the manuscript which of these comparison are exact. [critical]

Error bars are not included in any of the tables. I suspect that this is because of how expensive it is to train these networks. It is necessary however to understand whether the proposed model exhibits variance or not. Please provide error bars for at least the DWAViT-S entries in Table 2. [critical]

### Minor typos [recommendation]
Missing closing brackets in equation 6.

Double our in paragraph 2 of 4.2.

**Strengths And Weaknesses:**

The idea is well presented and evaluated on a range of downstream tasks: classification, detection and segmentation. The experiments are at scale and compare against several related prior models. Comparisons are made between the two angular attention functions and against dot-product attention (ablation). The model does not cause a major overhead in terms of #params and #flops.

---

> ### Author Response · Authors · 2023-09-16
> **Response one**
>
> **Q1: In page 3, the manuscript identifies Dosovitskiy et al. 2021 as the first introduction of self-attention to computer vision. The method that first introduced self attention to computer vision is https://arxiv.org/abs/1802.05751 or https://arxiv.org/pdf/1711.07971.pdf; and transformer inspired architectures have been since applied to classification, for example here. The idea of using image patches as tokens was also implemented in https://arxiv.org/abs/1911.03584.**
>
> **Several alternatives to dot-product attention were explored in "Exploring Self-attention for Image Recognition". The "Non local networks" paper also discusses some alternatives to dot-product attention. These and similar works that appeared prior to ViT need to be cited.**
>
> **R1:** Thanks for your valuable suggestion. We have cited and discussed the suggested references in Section 2.
> The added discussions are as follows:
>
> The pioneering work [1, 2] first introduced the self-attention mechanism to the computer vision field, and some early work [3,4]  applied self-attention in the computer vision tasks.
>
> Early work [2,5] explored the general form of the function in the self-attention and proposed several operations such as dot-product and concatenation.
>
> [1] Parmar, Niki, et al. "Image transformer." International conference on machine learning. PMLR, 2018.
>
> [2] Wang, Xiaolong, et al. "Non-local neural networks." Proceedings of the IEEE conference on computer vision and pattern recognition. 2018.
>
> [3] Ramachandran, Prajit, et al. "Stand-alone self-attention in vision models." Advances in neural information processing systems 32 (2019).
>
> [4] Cordonnier, Jean-Baptiste, Andreas Loukas, and Martin Jaggi. "On the Relationship between Self-Attention and Convolutional Layers." International Conference on Learning Representations. 2019.
>
> [5] Zhao, Hengshuang, Jiaya Jia, and Vladlen Koltun. "Exploring self-attention for image recognition." Proceedings of the IEEE/CVF conference on computer vision and pattern recognition. 2020.
>
> **Q2: The analysis quantifies performance when controlling for #params and #flops. Please also quote the actual runtime in training steps per second for a fixed hardware setup (#gpus) used across all experiments for at least some subset of runs. In many cases, although the #flops are fixed, the actual runtime can blow up because of subtle differences in the formulation.**
>
> **R2:** We have added the runtime results of our model and Swin Transformer [1] in our manuscript. The table below shows the runtime of our model and the Swin Transformer [1] in image classification task. The experiment is implemented on a single A100 GPU. For fair comparisons, we fix the batch size to 100 for all the models. We report the average time per iteration in the table. The results show that our model would take the time twice as much as that of Swin Transformer due to the different self-attention function and the different local window mechansim. But the actual runtime doesn’t blow up when the size of the model scales up. More results can be found in Table 12 and Table 13 of our revised paper.
>
> [1] Ze Liu, Yutong Lin, Yue Cao, Han Hu, Yixuan Wei, Zheng Zhang, Stephen Lin, and Baining Guo. Swin transformer: Hierarchical vision transformer using shifted windows. In Proceedings of the IEEE/CVF international conference on computer vision, pp. 10012–10022, 2021.
>
> |  Model                |   #Param(M)  |  #FLOPs(G)   | time_per_iteration(s)   |
> |  :----  | :----: | :----:  | :----: |
> | Swin-T                  |      28.3         |        4.5          |            0.15                  |
> | DWAViT-T[cos]     |      22.7         |        4.2          |            0.29                  |
> | DWAViT-T[quad]   |      22.7         |        4.2          |            0.31                  |
> | Swin-S                  |      49.6         |        8,7          |            0.23                 |
> | DWAViT-S[cos]     |      44.6         |        8.2          |            0.43                  |
> | DWAViT-S[quad]   |      44.6         |        8.2          |            0.38                  |
> | Swin-B                  |      87.8         |        15.4         |           0.30                  |
> | DWAViT-B[cos]     |      77.4         |        14.3          |            0.60                  |
> | DWAViT-B[quad]   |      77.4         |        14.3          |            0.64                  |

---

> > ### Author Response · Authors · 2023-09-16
> > **Response two**
> >
> > **Q3:The analysis mainly compares against several baselines but it is not clear which baselines are actually an exact comparison. Exact comparison = same prediction head + same training regime + same pre-training and fine-tuning datasets. Please clarify in the manuscript which of these comparisons are exact.**
> >
> > **R3:**  Thanks for your suggestion. The exact comparison for each task is provided as follows. We have also added these clarifications on exact comparisons in our manuscript.
> >
> > Image classification: DeiT [1], CvT [2], Swin Transformer [3], Focal Transformer [4], DaViT [5], MViTv2 [6], CSWin Transformer [7] are baselines for the exact comparison. These baselines and our model are trained on ImageNet-1K from scratch on A100 with the same training recipe and we report the accuracy of these models in Table 2.
> >
> > Object detection: In Table 3 and Table 4, Swin Transformer [3], Focal Transformer [4], XciT [9], DAT [8], CSWin Transformer [7] are baselines for the exact comparison. For these baselines and our model, we adopt MMDetection toolbox to perform the object detection task on COCO. We adopt the models pre-trained on ImageNet-1K only as the feature extractor and Mask R-CNN [10] as the prediction head.
> >
> > In Table 5, Swin Transformer [3] and DAT [8] are baselines for the exact comparison.  For these baselines and our model, we adopt MMDetection toolbox to perform the object detection task on COCO. We adopt the models pre-trained on ImageNet-1K only as the feature extractor and  Cascade Mask R-CNN [11] as the prediction head.
> >
> > Semantic segmentation: In Table 6 and Table 14, Swin Transformer [3], Focal Transformer [4], XciT [9], DaViT [5], DAT [8]  are baselines for the exact comparison. For these baselines and our model, we adopt MMSegmentation toolbox to perform the semantic segmentation task on ADE20K. We adopt the models pre-trained on ImageNet-1K only as the feature extractor and Upernet [12] as the prediction head.
> >
> >
> > [1] Hugo Touvron, Matthieu Cord, Matthijs Douze, Francisco Massa, Alexandre Sablayrolles, and Hervé Jégou. Training data-efficient image transformers & distillation through attention. In International conference on machine learning, pp. 10347–10357. PMLR, 2021a.
> >
> > [2] Haiping Wu, Bin Xiao, Noel Codella, Mengchen Liu, Xiyang Dai, Lu Yuan, and Lei Zhang. Cvt: Introducing convolutions to vision transformers. In Proceedings of the IEEE/CVF International Conference on Computer Vision, pp. 22–31, 2021.
> >
> > [3] Ze Liu, Yutong Lin, Yue Cao, Han Hu, Yixuan Wei, Zheng Zhang, Stephen Lin, and Baining Guo. Swin transformer: Hierarchical vision transformer using shifted windows. In Proceedings of the IEEE/CVF international conference on computer vision, pp. 10012–10022, 2021.
> >
> > [4] Jianwei Yang, Chunyuan Li, Pengchuan Zhang, Xiyang Dai, Bin Xiao, Lu Yuan, and Jianfeng Gao. Focal attention for long-range interactions in vision transformers. In NeurIPS, pp. 30008–30022, 2021.
> >
> > [5] Mingyu Ding, Bin Xiao, Noel Codella, Ping Luo, Jingdong Wang, and Lu Yuan. Davit: Dual attention vision transformers. In Computer Vision–ECCV 2022: 17th European Conference, Tel Aviv, Israel, October 23–27, 2022, Proceedings, Part XXIV, pp. 74–92. Springer, 2022.
> >
> > [6] Yanghao Li, Chao-Yuan Wu, Haoqi Fan, Karttikeya Mangalam, Bo Xiong, Jitendra Malik, and Christoph Feichtenhofer. Mvitv2: Improved multiscale vision transformers for classification and detection. In Proceedings of the IEEE/CVF Conference on Computer Vision and Pattern Recognition, pp. 4804–4814, 2022.
> >
> > [7] Xiaoyi Dong, Jianmin Bao, Dongdong Chen, Weiming Zhang, Nenghai Yu, Lu Yuan, Dong Chen, and Baining Guo. Cswin transformer: A general vision transformer backbone with cross-shaped windows. In Proceedings of the IEEE/CVF Conference on Computer Vision and Pattern Recognition, pp. 12124–12134, 2022.
> >
> > [8] Zhuofan Xia, Xuran Pan, Shiji Song, Li Erran Li, and Gao Huang. Vision transformer with deformable attention. In Proceedings of the IEEE/CVF conference on computer vision and pattern recognition, pp.4794–4803, 2022.
> >
> > [9] Alaaeldin Ali, Hugo Touvron, Mathilde Caron, Piotr Bojanowski, Matthijs Douze, Armand Joulin, Ivan Laptev, Natalia Neverova, Gabriel Synnaeve, Jakob Verbeek, et al. Xcit: Cross-covariance image transformers. Advances in neural information processing systems, 34:20014–20027, 2021.
> >
> > [10] Kaiming He, Georgia Gkioxari, Piotr Dollar, and Ross Girshick. Mask r-cnn. In Proceedings of the IEEE International Conference on Computer Vision (ICCV), Oct 2017.
> >
> > [11] Zhaowei Cai and Nuno Vasconcelos. Cascade r-cnn: Delving into high quality object detection. In Proceedings of the IEEE Conference on Computer Vision and Pattern Recognition (CVPR), June 2018.
> >
> > [12] Tete Xiao, Yingcheng Liu, Bolei Zhou, Yuning Jiang, and Jian Sun. Unified perceptual parsing for scene understanding. In Proceedings of the European conference on computer vision (ECCV), pp. 418–434, 2018.

---

> > > ### Author Response · Authors · 2023-09-16
> > > **Response three**
> > >
> > > **Q4: Error bars are not included in any of the tables. I suspect that this is because of how expensive it is to train these networks. It is necessary however to understand whether the proposed model exhibits variance or not. Please provide error bars for at least the DWAViT-S entries in Table 2.**
> > >
> > > **R4:**  Thanks for your suggestion. We do not include the error bars in the tables for two reasons. First, in the vision transformer literature [1-9], the error bar is usually not included in the results of the classification task. For vision transformers, the model is trained on the large dataset (ImageNet-1K) for many training epochs (300 epochs). As a result, the variance could be diminished considerably and become insignificant compared to the main results. Thus, it is common to not include the error bar in the results. Second, we run our tiny-sized DWAViT model several times and find the variance is within 0.1% for the classification task. As a result, we do not include the error bar in the table.
> > >
> > > [1] Hugo Touvron, Matthieu Cord, Matthijs Douze, Francisco Massa, Alexandre Sablayrolles, and Hervé Jégou. Training data-efficient image transformers & distillation through attention. In International conference on machine learning, pp. 10347–10357. PMLR, 2021a.
> > >
> > > [2] Haiping Wu, Bin Xiao, Noel Codella, Mengchen Liu, Xiyang Dai, Lu Yuan, and Lei Zhang. Cvt: Introducing convolutions to vision transformers. In Proceedings of the IEEE/CVF International Conference on Computer Vision, pp. 22–31, 2021.
> > >
> > > [3] Ze Liu, Yutong Lin, Yue Cao, Han Hu, Yixuan Wei, Zheng Zhang, Stephen Lin, and Baining Guo. Swin transformer: Hierarchical vision transformer using shifted windows. In Proceedings of the IEEE/CVF international conference on computer vision, pp. 10012–10022, 2021.
> > >
> > > [4] Jianwei Yang, Chunyuan Li, Pengchuan Zhang, Xiyang Dai, Bin Xiao, Lu Yuan, and Jianfeng Gao. Focal attention for long-range interactions in vision transformers. In NeurIPS, pp. 30008–30022, 2021.
> > >
> > > [5] Mingyu Ding, Bin Xiao, Noel Codella, Ping Luo, Jingdong Wang, and Lu Yuan. Davit: Dual attention vision transformers. In Computer Vision–ECCV 2022: 17th European Conference, Tel Aviv, Israel, October 23–27, 2022, Proceedings, Part XXIV, pp. 74–92. Springer, 2022.
> > >
> > > [6] Yanghao Li, Chao-Yuan Wu, Haoqi Fan, Karttikeya Mangalam, Bo Xiong, Jitendra Malik, and Christoph Feichtenhofer. Mvitv2: Improved multiscale vision transformers for classification and detection. In Proceedings of the IEEE/CVF Conference on Computer Vision and Pattern Recognition, pp. 4804–4814, 2022.
> > >
> > > [7] Xiaoyi Dong, Jianmin Bao, Dongdong Chen, Weiming Zhang, Nenghai Yu, Lu Yuan, Dong Chen, and Baining Guo. Cswin transformer: A general vision transformer backbone with cross-shaped windows. In Proceedings of the IEEE/CVF Conference on Computer Vision and Pattern Recognition, pp. 12124–12134, 2022.
> > >
> > > [8] Zhuofan Xia, Xuran Pan, Shiji Song, Li Erran Li, and Gao Huang. Vision transformer with deformable attention. In Proceedings of the IEEE/CVF conference on computer vision and pattern recognition, pp.4794–4803, 2022.
> > >
> > > [9] Alaaeldin Ali, Hugo Touvron, Mathilde Caron, Piotr Bojanowski, Matthijs Douze, Armand Joulin, Ivan Laptev, Natalia Neverova, Gabriel Synnaeve, Jakob Verbeek, et al. Xcit: Cross-covariance image transformers. Advances in neural information processing systems, 34:20014–20027, 2021.
> > >
> > > **Q5: Computer vision, especially object detection, is an ethically sensitive topic. A broader impact statement is not present and I strongly encourage the authors to add one.**
> > >
> > > **R5:** Thank you for your suggestion. We have added a broader impact statement at the end of our manuscript. Our broader impact statement is as follows:
> > >
> > > This work proposed a new architecture of vision transformer called DWAViT featured by angular self-attention and dual local window mechanism. Our model is proven to achieve competitive performance in downstream tasks such as object detection and semantic segmentation and has the enormous potential to be used in various practical scenarios. In particular, object detection is one of the most promising applications of vision transformers in the real world and it is often used in systems which require extensive interaction with the surrounding environment visually. For instance,  autonomous vehicles require a large number of object detectors to identify the pedestrians and other vehicles nearby. Therefore, the safety and the trustworthiness of vision transformers are critical in this area. Though our proposed model can achieve promising results on object detection and other tasks, some critical issues such as adversarial robustness and trustworthiness are quite under-explored and further investigation is necessarily required.
> > >
> > > **Q6: Minor typos:
> > > Missing closing brackets in equation 6.
> > > Double our in paragraph 2 of 4.2.**
> > >
> > > **Q6:** Thanks for your suggestion. We have modified the typos in our manuscript.

---

### Decision · Action_Editor_4Ld4 · 2023-11-07

**Recommendation:** Reject

**Comment:**

This paper presented two techniques to improve the Swin Transformer for vision tasks, including dual window and angular self-attention to replace the corresponding original designs of Swin Transformer. The proposed models were tested on a series of tasks, including ImageNet classification, COCO detection and instance segmentation, and ADE20k segmentation.

After the rebuttal, two reviewers rated the paper "Accept" and 1 reviewer rated the paper "Leaning to Reject". The main negative comment is on the slow inference speed of the proposed angular self-attention mechanism, which the AE agrees. If a network is as twice slow as other ViT models, it is quite unlikely any follow-up work will choose the DWAViT as the backbone, as nowadays experiments are already quite slow. The authors shall carefully discuss and/or consider how to solve the slow inference problem (not just simply admit it is twice slow and move on).

In addition, the AE found that the paper lacks adequate ablation study on the proposed dual window design. Most ablation study focuses on the angular attention design. As the paper also claims the superiority of the dual window design, this component should also be carefully studied (better on multiple datasets and/or tasks).

**Audience:**

Yes.

**Claims And Evidence:**

Some claims lack strong evidence support.

1) The effectiveness of the proposed component should be proven under (mostly) similar parameters, FLOPs, and running time with compared models. However, the proposed DWAViT is currently twice as slow as the Swin Transformer. The authors lack discussion on the possible cause and feasible solutions to solve the running time issue.

2) The AE found that the paper lacks adequate ablation study on the proposed dual window design. Most ablation study focuses on the angular attention design. As the paper also claims the superiority of the dual window design, this component should also be carefully studied (better on multiple datasets and/or tasks).

**Resubmission Of Major Revision:**

The authors may consider submitting a major revision at a later time.